# A Likelihood Based Approach to Distribution Regression Using Conditional Deep Generative Models

Shivam Kumar [1]   Yun Yang [2]   Lizhen Lin [2]

## Abstract

In this work, we explore the theoretical properties of conditional deep generative models under the statistical framework of distribution regression where the response variable lies in a high-dimensional ambient space but concentrates around a potentially lower-dimensional manifold. More specifically, we study the large-sample properties of a likelihood-based approach for estimating these models. Our results lead to the convergence rate of a sieve maximum likelihood estimator (MLE) for estimating the conditional distribution (and its devolved counterpart) of the response given predictors in the Hellinger (Wasserstein) metric. Our rates depend solely on the intrinsic dimension and smoothness of the true conditional distribution. These findings provide an explanation of why conditional deep generative models can circumvent the curse of dimensionality from the perspective of statistical foundations and demonstrate that they can learn a broader class of nearly singular conditional distributions. Our analysis also emphasizes the importance of introducing a small noise perturbation to the data when they are supported sufficiently close to a manifold. Finally, in our numerical studies, we demonstrate the effective implementation of the proposed approach using both synthetic and real-world datasets, which also provide complementary validation to our theoretical findings.

[1]Department of Applied and Computational Mathematics and Statistics, University of Notre Dame, Notre Dame, USA [2]Department of Mathematics, University of Maryland, College Park, USA. Correspondence to: Shivam Kumar <skumar9@nd.edu>.

*Proceedings of the 42nd International Conference on Machine Learning*, Vancouver, Canada. PMLR 267, 2025. Copyright 2025 by the author(s).

## 1. Introduction

Conditional distribution estimation provides a principled framework for characterizing the dependence relationship between a response variable $Y$ and predictors $X$, with the primary goal of estimating the distribution of $Y$ conditional on $X$ through learning the (conditional) data-generating process. Conditional distribution estimation allows one to regress the entire distribution of $Y$ on $X$, which provides much richer information than the traditional mean regression and plays a central role in various important areas ranging from causal inference (Pearl, 2009; Spirtes, 2010), graphical models (Jordan, 1999; Koller & Friedman, 2009), representation learning (Bengio et al., 2013), dimension reduction (Carreira-Perpinán, 1997; Van Der Maaten et al., 2009), to model selection (Claeskens & Hjort, 2008; Ando, 2010). Their applications span across diverse domains such as forecasting (Gneiting & Katzfuss, 2014), biology (Krishnaswamy et al., 2014), energy (Jeon & Taylor, 2012), astronomy (Zhao et al., 2021), and industrial engineering (Simar & Wilson, 2015), among others.

There is a rich literature in statistics and machine learning on conditional distribution estimation including both frequentist and Bayesian methods (Hall & Yao, 2005; Norets & Pati, 2017). Traditional methods, however, suffer from the curse of dimensionality and often struggle to adapt to the intricacies of modern data types such as the ones with lower-dimensional manifold structures.

Recent methodologies that leverage deep generative models have demonstrated significant advancements in complex data generation. Instead of explicitly modeling the data distribution, these approaches implicitly estimate it through learning the corresponding data sampling scheme. Commonly, these implicit distribution estimation approaches can be broadly categorized into three types. The first one is likelihood-based with notable examples including Kingma & Welling (2013), Rezende et al. (2014), Burda et al. (2015), and Song et al. (2021) . The second approach, based on adversarial learning, matches the empirical distribution of the data with a distribution estimator using an adversarial loss. Representative examples include Goodfellow et al. (2014), Arjovsky et al. (2017), and Mroueh et al. (2017), among others. The third approach, which is more recent, reduces

the problem of distribution estimation to score estimation through certain time-discrete or continuous dynamical systems. The idea of score matching was first proposed in Hyvärinen & Dayan (2005) and Vincent (2011). More recently, score-based diffusion models have achieved state-of-the-art performance in many applications (Sohl-Dickstein et al., 2015; Nichol & Dhariwal, 2021; Song et al., 2020; Lipman et al., 2022; Brehmer & Cranmer, 2020; Han et al., 2022).

On the theoretical front, recent works such as Liu et al. (2021), Chae et al. (2023), Altekrüger et al. (2023), Stanczuk et al. (2024), Pidstrigach (2022) , and Tang & Yang (2023) demonstrate that distribution estimation based on deep generative models can adapt to the intrinsic geometry of the data, with convergence rates dependent on the intrinsic dimension of the data, thus potentially circumventing the curse of dimensionality. Such advancement has naturally motivated us to employ and investigate *conditional deep generative model* for conditional distribution estimation. Specifically, we explore and study the theoretical properties of a new likelihood-based approach to conditional sampling using deep generative models for data potentially residing on a low-dimensional manifold corrupted by full-dimensional noise. More concretely, we consider the following *conditional distributional regression* problem:

$$Y|X = V|X + \varepsilon, \tag{1}$$

where $X$ serves as a predictor in $\mathbb{R}^{\mathfrak{p}}$, $V|X$ represents the (uncorrupted) underlying response supported on a manifold of dimension $\mathfrak{d} \leq D$, $Y|X$ represents the observed response, and $\varepsilon \sim \mathsf{N}(0, \sigma_*^2 I_D)$ denotes the noise residing in the ambient space $\mathbb{R}^D$. Our deep generative model focuses on the conditional distribution $V|X$ by using a (conditional) generator of the form $G_*(Z, X)$, where $G_*$ is a function of a random seed $Z$ and the covariate information $X$. This approach is termed 'conditional deep generative' because the conditional generator is modeled using deep neural networks (DNNs). Observe that, when $\mathfrak{d} < D$, the distribution of $G_*(Z, X)$ is supported on a lower-dimensional manifold, making it singular with respect to the Lebesgue measure in the $D$-dimensional ambient space. We study the statistical convergence rate of sieve MLEs in the conditional deep general model setup and investigate its dependence on the intrinsic dimension, structure properties of the model as well as the noise level of the data.

## 1.1. List of contributions

We briefly summarise the main contributions made in this paper.

- To the best of our knowledge, our study is the first attempt to explore the likelihood-based approach for distributional regression using a conditional deep generative

model, considering full-dimensional noise and the potential presence of singular underlying support. We provide a solid statistical foundation for the approach by proving the near-optimal convergence rates for this proposed estimator.

- We derive the convergence rates for the conditional density estimator of the corrupted data $Y$ with respect to the Hellinger distance and specialize the obtained rate for two popular deep neural network classes: the sparse and fully connected network classes. Furthermore, we characterize the Wasserstein convergence rates for the induced intrinsic conditional distribution estimator on the manifold (i.e., a deconvolution problem). Both rates turn out to depend only on the intrinsic dimension and smoothness of the true conditional distribution.

- Our analysis in Corollary 2 suggests the need to inject a small amount of noise into the data when they are sufficiently close to the manifold. Intuitively, this observation validates the underlying structural challenges in related manifold estimation problems with noisy data, as outlined by Genovese et al. (2012).

- We show that the class of learnable (conditional) distributions of our method is broad. It encompasses not only the smooth distributions class, but also extends to the general (nearly) singular distributions with manifold structures, with minimal assumptions.

## 1.2. Other relevant literature

The problem of non-parametric conditional density estimation has been extensively explored in statistical literature. Hall & Yao (2005), Bott & Kohler (2017), and Bilodeau et al. (2023) directly tackle this problem with smoothing and local polynomial-based methods. Fan & Yim (2004) and Efromovich (2007) explore suitably transformed regression problems to address this challenge. Other notable approaches include the nearest neighbor method (Izbicki et al., 2020; Bhattacharya & Gangopadhyay, 1990), basis function expansion (Sugiyama et al., 2010; Izbicki & Lee, 2016), tree-based boosting (Pospisil & Lee, 2018; Gao & Hastie, 2022), and Bayesian optimal transport flow Chemseddine et al. (2024) among others.

In the context of conditional generation, we highlight recent work by Zhou et al. (2022) and Liu et al. (2021). In Zhou et al. (2022), GANs were employed to investigate conditional density estimation. While this work offers a consistent estimator, it lacks statistical rates or convergence analysis, and its focus is on a low-dimensional setup. In Liu et al. (2021), conditional density estimation supported on a manifold using Wasserstein-GANs was examined. However, their setup does not account for smoothness across either covariates or responses, nor do they address how deep generative models specifically tackle the challenges of high-

dimensionality. Moreover, their assumption that the data lies exactly on the manifold can be restrictive. Our study shares some commonalities with the work of Chae et al. (2023), as both investigate sieve maximum likelihood estimators (MLEs). However, the fundamental problems addressed and the methodologies employed differ significantly, and our work involves technical challenges that span multiple scales. While Chae et al. (2023) concentrates exclusively on unconditional distribution estimation, our theoretical analysis necessitates much more nuanced techniques due to the conditional nature of our setup. This shift is noteworthy because it demands a more refined analysis of entropy bounds, considering two potential sources of smoothness - across the regressor and the response variables. Furthermore, our setting accommodates the possibility of an infinite number of $x$ values, which gives rise to a dynamic manifold structure, further compounding the intricacy of the problem at hand.

## 2. Conditional deep generative models for distribution regression

We consider the following probabilistic conditional generative model, where for a given predictor value $x$, the response $Y$ is generated by

$$Y = G_*(Z, x) + \varepsilon, \quad x \in \mathcal{X} \subset \mathbb{R}^{\mathfrak{p}}. \quad (2)$$

Here, $G_*(\cdot, x) : \mathcal{Z} \to \mathcal{M}_x$ is the unknown generator function, $Z$ a latent variable with a known distribution $P_Z$ and support $\mathcal{Z} \subset \mathbb{R}^{\mathfrak{d}}$ independent of the predictor $X$. The existence of the generator $G_*$ directly follows from Noise Outsourcing Lemma 3. This lemma enables the transfer of randomness into the covariate and an orthogonal (independent) component through a generating function for any regression response. We denote $\mathcal{M} := \cup_{x \in \mathcal{X}} \mathcal{M}_x \subset \mathbb{R}^D$ as the support of the image of $G_*(\mathcal{Z}, \mathcal{X})$ such as a (union of) $d$-dimensional manifold. We model $G_*(\cdot, \cdot) : \mathcal{Z} \times \mathcal{X} \subset \mathbb{R}^{\mathfrak{d}} \times \mathbb{R}^{\mathfrak{p}} \to \mathcal{Y} \subset \mathbb{R}^D$ using a deep neural network, leading to a *conditional deep generative model* for (2).

In the next section, we present a more general result in terms of the entropy bound (variance) for the true function class of $G_*$ and the approximability (bias) of the search class. We then proceed to a simplified understanding in the context of conditional deep generative models in subsequent sections.

### 2.1. Convergence rates of the Sieve MLE

In light of equation (2), it is evident that the distribution of $Y|X = x$ results from the convolution of two distinct distributions: the pushforward of $Z$ through $G_*$ with $X = x$, and $\varepsilon$ following an independent $D$-dimensional normal distribution. The density corresponding to the true distribution $P_*(\cdot|X = x)$ can thus be expressed as:

$$p_*(y|x) = \int \phi_{\sigma_*}(y - G_*(z, x)) \, dP_Z,$$

where $\phi_{\sigma_*}$ is the density of $\mathsf{N}(0, \sigma_*^2 I_d)$. We define the class of conditional distributions $\mathcal{P}$ as

$$\mathcal{P} = \left\{ P_{g,\sigma} : g(\cdot, x) \in \mathcal{F}, \sigma \in [\sigma_{\min}, \sigma_{\max}] \right\}, \quad (3)$$

where $P_{g,\sigma}$ represents the distribution with density $p_{g,\sigma} = \int \phi_\sigma(y - g(z, x)) dP_Z$. In this notation, $P_* = P_{G_*, \sigma_*}$ and $p_* = p_{G_*, \sigma_*}$. The elements of $\mathcal{P}$ comprise two components: $g$ originating from the underlying function class $\mathcal{F}$, and $\sigma$, which characterizes the noise component. This class enables us to obtain separate estimates for $G_*$ and $\sigma_*$, furnishing us with both the canonical estimator for the distribution of $Y|X = x$ and enhancing our comprehension of the singular distribution of $G_*(Z, x)$, supported on a low-dimensional manifold.

Given a data set $\{(X_i, Y_i)\}_{i=1}^n$, the log-likelihood function is defined as $\ell_n(g, \sigma) = n^{-1} \sum_{i=1}^n \log p_{g,\sigma}(Y_i|X_i)$. For a sequence $\eta_n \downarrow 0$ as $n \to \infty$, a *sieve* maximum likelihood estimator (MLE) (Geman & Hwang, 1982) is any estimator $(\widehat{g}, \widehat{\sigma}) \in \mathcal{F} \times [\sigma_{\min}, \sigma_{\max}]$ that satisfies

$$\ell_n(\widehat{g}, \widehat{\sigma}) \geq \sup_{\substack{\sigma \in [\sigma_{\min}, \sigma_{\max}] \\ g \in \mathcal{F}}} \ell_n(g, \sigma) - \eta_n. \quad (4)$$

Here $\widehat{g} \in \mathcal{F}$ and $\widehat{\sigma} \in [\sigma_{\min}, \sigma_{\max}]$ are the estimators, and $\eta_n$ represents the optimization error. The dependence of $\widehat{g}$ and $\widehat{\sigma}$ on $n$ illustrates the sieve's role in approximating the true distribution when optimization is performed over the class $\mathcal{P}$. The estimated density $\widehat{p} = p_{\widehat{g}, \widehat{\sigma}}$ provides an estimator for $p_*(\cdot|\cdot)$, and $Q_{\widehat{g}}(\cdot|X = x)$ serve as the estimator for $Q_*(\cdot|X = x)$.

In this section, we formulate the main results, which provide convergence rates in the Hellinger distance for our sieve MLE estimator. The convergence rate was derived for any search functional class $\mathcal{F}$, with a brief emphasis on their entropy and approximation capabilities.

**Assumption 1** (True distribution). *Denote $\mu_X^*(x)$ as the distribution of $X$. We denote the true conditional densities as $p_* = \{p_*(\cdot|x), x \in \mathbb{R}^{\mathfrak{p}}\}$. It is natural to assume that the data is generated from $p_*$ from model (2) with some true generator $G_*$ and $\sigma_*$. We denote $Q_*(\cdot|X = x)$ (or $Q_{G_*}$) as the distribution of $G_*(Z, x)$ for some distribution $P_Z$.*

A function $g$ is said to have a composite structure (Schmidt-Hieber, 2020; Kohler & Langer, 2021) if it takes the form as

$$g = f_q \circ f_{q-1} \circ \cdots \circ f_1 \quad (5)$$

where $f_j : (a_j, b_j)^{d_j} \to (a_{j+1}, b_{j+1})^{d_{j+1}}$, $d_0 = \mathfrak{p} + \mathfrak{d}$ and $d_{q+1} = D$. Denote $f_j = (f_j^{(1)}, \ldots, f_j^{(d_{j+1})})$ as

the components of $f_j$, let $t_j$ be the maximal number of variables on which each of the $f_j^{(i)}$ depends and let $f_j^{(i)} \in \mathcal{H}^{\beta_j}((a_j, b_j)^{t_j}, K)$ (see Section 2.4.1 for the definition of the Hölder class $\mathcal{H}^\beta$). A composite structure is very general which includes smooth functions and additive structure as special cases. In addition, in the next section, we show the class of conditional distributions $\{Q_{G_*}(\cdot|X=x) : x \in \mathbb{R}^{\mathfrak{p}}, G_* \in \mathcal{G}\}$ induced by the composite structure is broad.

**Assumption 2** (composite structure). *Denote $\mathcal{G} = \mathcal{G}(q, \boldsymbol{d}, \boldsymbol{t}, \boldsymbol{\beta}, K)$ as a collection of functions of form (5), where $\boldsymbol{d} = (d_0, \ldots, d_{q+1})$, $\boldsymbol{t} = (t_0, \ldots, t_{q+1})$, and $\boldsymbol{\beta} = (\beta_0, \ldots, \beta_{q+1})$. We regard $(q, \boldsymbol{d}, \boldsymbol{t}, \boldsymbol{\beta}, K)$ as constants in our setup, and assume that the true generator $G_*(\cdot, x)$ as in (2) belongs to $\mathcal{G}$, for all $x \in \mathcal{X}$. Additionally, we assume $\||G_*|_\infty\|_\infty \leq K$.*

$$\widetilde{\beta}_j = \beta_j \prod_{l=j+1}^{q} (\beta_l \wedge 1), \quad j_* = \underset{j \in \{0, \ldots, q\}}{\operatorname{argmax}} \frac{t_j}{\widetilde{\beta}_j},$$

$$\beta_* = \widetilde{\beta}_{j_*}, \quad t_* = t_{j_*}.$$

*The quantities $t_*$ and $\beta_*$ are called intrinsic dimension and smoothness of $G_*$ (or of $\mathcal{G}$).*

**Remark 1** (Strength of the Composite Structure). *The expression $(a_j, b_j) \subset [-K, K]$ can be intuitively visualized by setting $a_j = -K$ and $b_j = K$. To illustrate the impact of intrinsic dimensionality and smoothness, consider a function $f : \mathbb{R}^d \to \mathbb{R}$ defined as $f(x) = f_1(x_1) + \ldots + f_d(x_d)$, where $x = (x_1, \ldots, x_d)$ and $f_j \in \mathcal{H}^\beta((-K, K), K)$ for $j = 1, \ldots, d$. While $f \in \mathcal{H}^\beta((-K, K)^d, K)$, its intrinsic dimension is $t_* = 1$ with intrinsic smoothness $\beta$. This mitigates the curse of dimensionality.*

**Example** (One-dimensional $\beta$-Hölder Generator). *Let $U \sim \mathrm{Unif}(0, 1)$ and define $G(u) = u^{1/\beta}, u \in [0, 1]$. Then $X = G(U)$ has density*

$$\frac{d}{dx}\mathbb{P}(U \leq x^\beta) = \beta \, x^{\beta-1}, \quad x \in [0, 1],$$

*which belongs to the $\beta$-Hölder class on $[0, 1]$. In our notation one checks $t_* = 1, \beta_* = \beta$, and setting $\beta = 1$ recovers $\mathrm{Unif}(0, 1)$ case and thus provides a fully explicit illustration of Assumption 2.*

**Assumption 3.** *Let $\mathcal{M}_*$ be the closure of $G_*(\mathcal{Z}, \mathcal{X})$. We assume that $\mathcal{M}_*$ does not have an interior point, and $\mathrm{reach}(\mathcal{M}_*) = \mathsf{r}_*$ with $\mathsf{r}_* > 0$.*

Assumption 2 permits low intrinsic dimensionality within the learnable function class. Assumption 3 imposes the strong identifiability condition necessary for efficient estimation, as seen in manifold literature (Aamari & Levrard, 2019; Tang & Yang, 2023).

Given two conditional densities $p_1(\cdot|x), p_2(\cdot|x)$ and $\mu_X^*$ denoting the density of $X$, we use integrated distances for a measure of evaluation. With a slight abuse of notation, we denote $d_1(p_1, p_2) = \mathbb{E}_X[d_1(p_1(\cdot|x), p_2(\cdot|x))]$ and $d_H(p_1, p_2) = \mathbb{E}_X[d_H(p_1(\cdot|x), p_2(\cdot|x))]$, where $d_1$ and $d_H$ represent the $L_1$ and the Hellinger distance as $d_1(p_1(\cdot|x), p_2(\cdot|x)) = \int |p_1(y|x) - p_2(y|x)| \, dy$ and $d_H(p_1, p_2) = (\int \int [\sqrt{p_1(y|x)} - \sqrt{p_2(y|x)}]^2 \, dy)^{1/2}$ respectively. Denote $\mathcal{N}(\delta, \mathcal{F}, d)$ and $\mathcal{N}_{[]}(\delta, \mathcal{F}, d)$ as covering and bracketing numbers of the function class $\mathcal{F}$ with respect to the (pseudo)-metric $d$.

We first present Lemma 1, which establishes the bracketing entropy of the functional class $\mathcal{P}$ with respect to Hellinger distance in terms of the covering entropy of the search class $\mathcal{F}$. This enables us to transfer the entropy control of the individual components $\mathcal{F}$ and $\sigma$ to the entire $\mathcal{P}$.

**Lemma 1.** *Let $\mathcal{F}$ be class of functions from $\mathcal{Z} \times \mathcal{X}$ to $\mathbb{R}^D$ such that $\||g|_\infty\|_\infty \leq K$ for every $g \in \mathcal{F}$. Let $\mathcal{P} = \{P_{g,\sigma} : g \in \mathcal{F}, \sigma \in [\sigma_{\min}, \sigma_{\max}]\}$ with $\sigma_{\min} \leq 1$. Then, there exist constants $c = c(\sigma_{\max}, K, D)$ and $C = C(\sigma_{\max}, K, D)$ and $\delta_* = \delta_*(D)$ such that for every $\delta \in (0, \delta_*]$,*

$$\log \mathcal{N}_{[]}(\delta, \mathcal{P}, d_H) \leq \log \mathcal{N}(c\sigma_{\min}^{D+3}\delta^4, \mathcal{F}, \|| \cdot |_\infty\|_\infty) + \log \left(\frac{C}{\sigma_{\min}^{D+2}\delta^4}\right), \quad (6)$$

The proof of Lemma 1 is provided in the Appendix E. Theorem 1 presents the convergence rate of the sieve-MLE to the true distribution (see Appendix F for the proof).

**Theorem 1.** *Let $\mathcal{F}, \mathcal{P}, \sigma_{\min}$ and $\delta_* = \delta_*(D)$ be given as in Lemma 1, and $n \geq 1$. Suppose that $\log \mathcal{N}(\delta, \mathcal{F}, \|| \cdot |_\infty\|_\infty) \leq \xi \{A + 1 \vee \log \delta^{-1}\}$ for every $\delta \in (0, \delta_*]$ and some $A, \xi > 0$. Suppose that there exists a $G \in \mathcal{F}$ and some $\delta_{\mathrm{approx}} \in (0, \delta_*]$ such that $\||G - G_*|_\infty\|_\infty \leq \delta_{\mathrm{approx}}$. Furthermore, suppose that $s \geq 1$, $A \geq 1$, $\sigma_{min} \leq 1$, $\delta_{\mathrm{approx}} \leq 1$ and $\sigma_* \in [\sigma_{\min}, \sigma_{\max}]$. Then*

$$P_*\left(d_H(\widehat{p}, p_*) > \varepsilon_n^*\right) \leq 5e^{-C_1 n \varepsilon_n^{*2}} + C_2 n^{-1} \quad (7)$$

*provided that $\eta_n \leq n\varepsilon_n^{*2}/6$ and $\varepsilon_n^* \leq \sqrt{2}\delta_*$, where*

$$\varepsilon_n^* = C_3 \left(\sqrt{\frac{\xi\{A + \log(n/\sigma_{\min})\}}{n}} \vee \frac{\delta_{\mathrm{approx}}}{\sigma_*}\right), \quad (8)$$

*$C_1$ is an absolute constant, $C_2 = C_2(D)$ and $C_3 = C_3(D, K, \sigma_{\max})$.*

The outlined rate has two components: the statistical component, expressed as an upper bound to the metric entropy of $\mathcal{F}$, and the approximation component, denoted as $\delta_{\mathrm{approx}}$. The statistical error is quantified by measuring the complexity of the class $\mathcal{P}$, as formulated in Lemma 1. The approximation error is assessed through the ability of the provided function class to approximate the true distribution.

## 2.2. Neural network class

We model $G_*(\cdot, \cdot)$ using a deep neural network. More specifically, we parameterize the true generator $G_*$ with a deep neural neural architecture $(L, \mathbf{r})$ of the form

$$f : \mathbb{R}^{r_0} \to \mathbb{R}^{r_{L+1}}, \quad z \mapsto f(z) = W_L \rho_{v_L} W_{L-1} \rho_{v_{L-}} \dots W_1 \rho_{v_1} W_0 z, \quad (9)$$

where $W_j \in \mathbb{R}^{r_{j+1} \times r_j}, v_j \in \mathbb{R}^{r_j}, \rho_{v_j}(\cdot) = \mathrm{ReLU}(\cdot - v_j)$ and $\mathbf{r} = (r_0, \dots, r_{L+1}) \in \mathbb{N}^{L+2}$. The constant $L$ is the number of hidden layers and $r = (r_0, \dots, r_{L+1})$ represents the number of nodes in each layer.

We define the **sparse** neural architecture class $\mathcal{F}_s(L, \mathbf{r}, s, B, K)$ as set of functions of form (9) satisfying

$$\max_{0 \le j \le L} |W_j|_\infty \vee |v_j|_\infty \le B, \quad \sum_{j=1}^{L} |W_j|_0 + |v_j|_0 \le s, \quad \||f|_\infty\|_\infty \le K,$$

with $r_0 = \mathfrak{d} + \mathfrak{p}$ and $r_{L+1} = D$, where $|\cdot|_0$ and $|\cdot|_\infty$ stand for the $L^0$ and $L^\infty$ vector norms, and $\||f|_\infty\|_\infty = \sup_{x \in \mathbb{R}^{r_0}} \max_{i=1,\dots,D} |f_i(x)|$, $s$ is sparsity parameter and $K$ is functional bound.

The **fully connected** neural architecture class $\mathcal{F}_c = \mathcal{F}_c(L, \mathbf{r}, B, K)$ is set of functions of form (9) satisfying

$$\max_{0 \le j \le L} |W_j|_\infty \vee |v_j|_\infty \le B, \qquad \||f|_\infty\|_\infty \le K.$$

Both classes $\mathcal{F}_s$ and $\mathcal{F}_c$ for the deep generator will be considered in our analysis of the resulting sieve maximum likelihood estimator. We denote the corresponding sieve-MLE as $\widehat{p}_s$ and $\widehat{p}_c$, respectively. When we use $r$ instead of $\mathbf{r}$, it refers to $r_1 = \dots = r_L = r$ along with $r_0 = \mathfrak{d} + \mathfrak{p}$ and $r_{L+1} = D$.

We can simplify and visualize the result stated in Theorem 1 in both cases: when the sieve-MLE is obtained with optimization performed over the class $\mathcal{F}_s$ and $\mathcal{F}_c$. To fulfill the conditions stated in the Theorem 1, we need to establish entropy bounds for these function classes, $\mathcal{F}_s$ and $\mathcal{F}_c$, and gain insight into their approximation capabilities for the composite structure class described in Assumption 2.

For the sparse neural architecture class $\mathcal{F}_s(L, r, s, K)$, the entropy, formally stated as Proposition 1 in Ohn & Kim (2019), is bounded as follows.

$$\log \mathcal{N}(\delta, \mathcal{F}_s, \||\cdot|_\infty\|_\infty) \lesssim sL \{\log(BLr) + \log \delta^{-1}\}. \quad (10)$$

From an entropy perspective, the fully connected neural architecture class $\mathcal{F}_c(L, r, B, K)$ can be viewed as $\mathcal{F}_s$ without any sparsity constraint, meaning $s \asymp r^2 L$. Therefore, we have

$$\log \mathcal{N}(\delta, \mathcal{F}_c, \||\cdot|_\infty\|_\infty) \lesssim L^2 r^2 \{\log(BLr) + \log \delta^{-1}\}. \quad (11)$$

The approximation properties of the sparse and fully connected network are provided in Lemma 4.1 and Lemma 4.2 of the Appendix K, respectively.

Having established the essential components for $\mathcal{F}_c$ in (11) and Lemma 4.2, and for $\mathcal{F}_s$ in (10) and Lemma 4.1, respectively, we can simplify Theorem 1 and state Corollary 1.

**Corollary 1.** *Suppose that Assumptions 1 and 2 hold, and $\sigma_* \in [\sigma_{\min}, \sigma_{\max}]$ with $\sigma_{\min} \le 1$ and $\sigma_{\max} < \infty$. Moreover, assume that the noise $\sigma_*$ decays at rate $\alpha$, i.e., $\sigma_* \asymp n^{-\alpha}$, and $\sigma_{\min} = n^{-\gamma}$ for some $\gamma \ge \alpha \ge 0$. Then, for every $\delta_{\mathrm{approx}} \in [0, 1]$, the following holds:*

1. *Let $\mathcal{F}_s = \mathcal{F}_s(L, r, s, B, K)$ with $\delta_* = \delta_*(D)$ be as given in Lemma 1, and $L \asymp \log \delta_{\mathrm{approx}}^{-1}$, $r \asymp \delta_{\mathrm{approx}}^{-t_*/\beta_*}$, $s \asymp \delta_{\mathrm{approx}}^{-t_*/\beta_*} \log \delta_{\mathrm{approx}}^{-1}$, $B \asymp \delta_{\mathrm{approx}}^{-1}$. Then the sieve MLE $\widehat{p}_s$ satisfies (7) with $\varepsilon_n^*$ as in (8) with $\xi = \delta_{\mathrm{approx}}^{-t_*/\beta_*} \log^2(\delta_{\mathrm{approx}}^{-1})$ and $A = \log^2(\delta_{\mathrm{approx}}^{-1})$ provided that $\eta_n \le n \varepsilon_n^{*2}/6$ and $\varepsilon_n^* \le \sqrt{2} \delta_*$.*

2. *Let $\mathcal{F}_c = \mathcal{F}_c(L, r, B, K)$ with $\delta_* = \delta_*(D)$ be as given in Lemma 1, and $L \asymp \log \delta_{\mathrm{approx}}^{-1}$, $r \asymp \delta_{\mathrm{approx}}^{-t_*/2\beta_*}$, $B \asymp \delta_{\mathrm{approx}}^{-1}$. Then the sieve MLE $\widehat{p}_c$ satisfies (7) with $\varepsilon_n^*$ as in (8) with $\xi = \delta_{\mathrm{approx}}^{-t_*/\beta_*} \log^2(\delta_{\mathrm{approx}}^{-1})$ and $A = \log^2(\delta_{\mathrm{approx}}^{-1})$ provided that $\eta_n \le n \varepsilon_n^{*2}/6$ and $\varepsilon_n^* \le \sqrt{2} \delta_*$.*

*In particular, choosing $\delta_{\mathrm{approx}} := (\sigma_*^2/n)^{\beta_*/(2\beta_*+t_*)}$ minimizes $\varepsilon_n^* \asymp \sqrt{\xi \{A + \log(n/\sigma_{\min})\}/n} \vee \delta_{\mathrm{approx}}/\sigma_*$, and gives*

$$\varepsilon_n^* \asymp n^{-\frac{\beta_* - t_* \alpha}{2\beta_* + t_*}} \log^2(n). \quad (12)$$

**Remark 2.** *The convergence rate in (12) illustrates the influence of intrinsic dimensionality, smoothness, and noise level on the estimation process. Note that $\alpha$ is upper bounded as $\varepsilon_n^* \le \sqrt{2} \delta_*(D)$. For large values of $\alpha$, estimation of $G_*$ is inherent difficult as the data is very close on the singular support. To address this, a small noise injection, as described in Corollary 2, can smooth the estimation and ensure consistency.*

The proof of Corollary 1 is provided in Appendix G. For the composite structural class $\mathcal{G}$, the effective smoothness is denoted by $\beta_*$, and the dimension is $t_*$. This effectively mitigates the curse of dimensionality. The convergence rate at (12) also recovers the optimal rate when $q = 1$ and $\alpha = 0$, and there is a small lag of polynomial factor $t_* \alpha/(2\beta_* + t_*)$ when $\alpha > 0$ (Norets & Pati, 2017). This lag arises due to the presence of full-dimensional noise in the response observation $Y$. Note that when the noise is small, that is $\alpha$ is large, achieving a sharp estimation of $p_*$ requires an equally accurate estimate of $G_*$. This can be quite challenging.

Our practically tractable approach attempts to address this without initially estimating the singular support.

## 2.3. Wasserstein convergence of the intrinsic (conditional) distributions

Using Wasserstein distance as a metric for distributions $Q_g$ is meaningful due to their singularity in ambient space: when $\mathfrak{d} < D$, the conditional distribution is singular with respect to the Lebesgue measure on $\mathbb{R}^D$.

The integrated Wasserstein distance, for $r \geq 1$, between $P_1(\cdot|X)$ and $P_2(\cdot|X)$ is defined as

$$W_r(P_1, P_2) = \mathbb{E}_X \left[ \inf_{\beta \in \Gamma(P_1, P_2)} \left( \mathbb{E}_{(U_1, U_2) \sim \beta} \left[ |U_1 - U_2|_r^r \right] \right)^{1/r} \right],$$

where $\Gamma(P_1, P_2)$ is the set of all couplings between $P_1$ and $P_2$ that preserves the two marginals. The (dual) representation of this norm, $W_r(P_1, P_2) = \mathbb{E}_X \left[ \sup_{\|f\|_{\mathrm{Lip}_r} \leq 1} \left\{ \mathbb{E}_{P_1}[f] - \mathbb{E}_{P_2}[f] \right\} \right]$ (Villani et al., 2009) with $\| \cdot \|_{\mathrm{Lip}_r}$ denoting the $r$-Lipschitz norm, is particularly useful in our proofs.

**Theorem 2.** *Suppose that Assumption 3 holds. If $d_H(p_{g,\sigma}, p_*) \leq \varepsilon$ holds for some $\varepsilon \in [0,1]$ and some $p_{g,\sigma} \in \mathcal{P}$, then we have*

$$W_1(Q_g, Q_*) \leq C \left( \varepsilon + \sigma_* \sqrt{\log \varepsilon^{-1}} \right),$$

*where $C = C(D, K, \mathsf{r}_*)$ depends only on $(D, K, \mathsf{r}_*)$.*

The proof of Theorem 2 is provided in Appendix H. Theorem 2 guarantees that $W_1\left( \widehat{Q}_{\widehat{g}}, Q_* \right) \lesssim_{\log} d_H(\widehat{p}, p_*) + \sigma_*$, where $\lesssim_{\log}$ represents less than or equal to up to a logarithmic factor of $n$. Following from Corollary 1, the Wasserstein convergence rate, $n^{-(\beta_* - t_* \alpha)/(2\beta_* + t_*)} \log^2(n) \vee \sigma_* \log^{1/2}(n)$, comprises two components: the convergence rate in the Hellinger distance and the standard deviation of the true noise sequence. It is noteworthy that the first expression is influenced by the variance of noise by the factor $\alpha$. When $\alpha$ is very small, indicating that the data $Y_j$ lies very close to the manifold, the second expression $n^{-\alpha}$ in the overall rate dominates. Intuitively, this phenomenon arises from the underlying structural challenges in related manifold estimation problems with noisy data, as discussed by Genovese et al. (2012). To address this issue, we propose a data perturbation strategy by transforming the data $\{(Y_j, X_j)\}_{j=1}^n$ into $\{(\widetilde{Y}_j, X_j)\}_{j=1}^n$, where $\widetilde{Y}_j = Y_j + \epsilon_j$ and $\epsilon_j \sim \mathsf{N}\left( 0_D, n^{-\beta_*/(\beta_* + t_*)} I_D \right)$. The resulting estimation error bound is summarized below, whose proof is provided in Appendix I.

**Corollary 2.** *Suppose that Assumption 1, 2, and 3 hold, and $\sigma_* \in [\sigma_{\min}, \sigma_{\max}]$ with $\sigma_* = n^{-\alpha}$ and $\sigma_{\min} = n^{-\gamma}$ for some $0 \leq \alpha \leq \gamma$. Then for each of the network architecture classes (sparse and fully connected) with the network*

*parameters specified in Corollary 1, the sieve MLE $\widehat{p}_{per}$ and $\widehat{Q}_{per}$ based on the perturbed data $\{(\widetilde{Y}_j, X_j)\}_{j=1}^n$ satisfies*

$$P_* \left[ W_1 \left( \widehat{Q}_{per}, Q_* \right) \geq \left( \varepsilon_n^* + \sigma_* \sqrt{\log \left( (\varepsilon_n^*)^{-1} \right)} \right) \right] \lesssim 5 e^{-C_1 n \varepsilon_n^{*2}} + \frac{C_2}{n}$$

*where $\varepsilon_n^*$ can be chosen such that*

$$\varepsilon_n^* + \sigma_* \sqrt{\log((\varepsilon_n^*)^{-1})} \asymp \begin{cases} n^{-\frac{\beta_* - t_* \alpha}{2\beta_* + t_*}} \log^2(n), & \text{if } \alpha < \beta_*/\{2(\beta_* + t_*)\}, \\ n^{-\frac{\beta_*}{2(\beta_* + t_*)}} \log^2(n), & \text{otherwise.} \end{cases}$$
$$(13)$$

## 2.4. Characterization of the learnable distribution class

Section 2.2 focuses on the true generator $G_*$ within the class of functions with composite structures. In this subsection, we show that such a conditional distribution class achieved by the push-forward map $G_*$ is broad and includes many existing distribution classes for $Q_*$ as special cases.

### 2.4.1. SMOOTH CONDITIONAL DENSITY

For $\beta > 0$, let $\mathcal{H}^\beta(D, M)$ be the class of all $\beta$-Hölder functions $f : D \subset \mathbb{R}^{\mathfrak{d}} \to \mathbb{R}$ with $\beta$-Hölder norm bounded by $M > 0$. Let $\mathcal{H}^\beta(D) = \cup_{M>0} \mathcal{H}^\beta(D, M)$. See Appendix B for their formal definitions.

**Lemma 2.** *Suppose that (i) $\mathcal{Z} \times \mathcal{X}$ and $\mathcal{Y}$ are uniformly convex and (ii) $p_Z \in \mathcal{H}^{\beta_Z}(\mathcal{Z})$, $\mu_X^* \in \mathcal{H}^{\beta_X}(\mathcal{X})$ and $q_* \in \mathcal{H}^{\beta_Q}(\mathcal{Y})$ for some $\beta_Z, \beta_X, \beta_Q > 0$ and are bounded above and below. Then, there exists a map $g(\cdot, \cdot) : \mathcal{Z} \times \mathcal{X} \to \mathcal{Y}$ such that $Q_*(\cdot|\cdot) = Q_g$ and $g \in \mathcal{H}^{\beta_{\min}+1}(\mathcal{Z} \times \mathcal{X})$, where $\beta_{\min} = \min\{\beta_Z, \beta_X, \beta_Q\}$.*

Lemma 2 establishes that the learnable distribution class includes Hölder-smooth functions with smoothness parameter $\beta_{\min}$ and intrinsic dimension $\mathfrak{d}$. As a result, following Corollary 1, the convergence rate for density estimation is given by $\varepsilon_n^* \asymp n^{-(\beta_{\min}+1-\mathfrak{d}\alpha)/(2\beta_{\min}+2+\mathfrak{d})}$. A push-forward map is a transport map between two distributions. The well-established regularity theory of transport map in optimal transport is directly applicable here [see Villani et al. (2009) and Villani (2021)]. The proof of Lemma 2 is based on Theorem 12.50 of (Villani et al., 2009) and Caffarelli (1996), which establishes the regularity of this transport map and its existence follows from Brenier (1991). When $p_Z$ is selected as a well-behaved parametric distribution, the regularity of the transport map is determined by the smoothness of both $\mu_X^*$ and $Q_*$. For a more detailed discussion on this, please refer to Appendix C.

### 2.4.2. A BROADER CONDITIONAL DISTRIBUTION CLASS WITH SMOOTHNESS DISPARITY

In Appendix L, we present a novel approximation result for the function class exhibiting *smoothness disparity* in Theorem 5. This new result facilitates the study of theoretical properties of estimators when the generator $G_* \in$

$\mathcal{H}_{\mathfrak{d},\mathfrak{p}}^{\beta_Z,\beta_X}(\mathcal{Z},\mathcal{X},K)$. Note that such a function class defined in (16) in Appendix L is much broader compared to the smoothness class in Section 2.4.1 as $Z$ and $X$ do not have to be jointly smooth and it allows for smoothness disparity among them. The subsequent Theorem 3 combines our approximation result with (11) and enables us to specialize Theorem 1 to this class (see Appendix J for the proof).

**Theorem 3.** *Let $G_* \in \mathcal{H}_{\mathfrak{d},\mathfrak{p}}^{\beta_Z,\beta_X}(\mathcal{Z},\mathcal{X},K)$. Suppose that Assumption 1 holds and $\sigma_* \in [\sigma_{\min},\sigma_{\max}]$ with $\sigma_{\min} \leq 1$ and $\sigma_{\max} < \infty$. Moreover, we assume $\sigma_* \asymp n^{-\alpha}$, and $\sigma_{\min} = n^{-\gamma}$ for some $0 \leq \alpha \leq \gamma \leq (\beta_Z^{-1}\mathfrak{d} + \beta_X^{-1}\mathfrak{p})^{-1}$. Then, for every $\delta_{\text{approx}} \in [0,1]$, we have: Let $\mathcal{F}_s = \mathcal{F}_s(L,r,s,1,K)$ with $L \asymp \log \delta_{\text{approx}}^{-1}$, $r \asymp \delta_{\text{approx}}^{-(\beta_Z^{-1}\mathfrak{d}+\beta_X^{-1}\mathfrak{p})}$, $s \asymp \delta_{\text{approx}}^{-(\beta_Z^{-1}\mathfrak{d}+\beta_X^{-1}\mathfrak{p})}\log \delta_{\text{approx}}^{-1}$. Then the sieve MLE $\widehat{p}_s$ satisfies (7) with the rate outlined in (8) with $\xi = \delta_{\text{approx}}^{-(\beta_Z^{-1}\mathfrak{d}+\beta_X^{-1}\mathfrak{p})}\log^2 \delta_{\text{approx}}^{-1}$ and $A = \log^2 \delta_{\text{approx}}^{-1}$, provided that $\eta_n \leq n\varepsilon_n^{*2}/6$. In particular, choosing $\delta_{\text{approx}} := \left(\sigma_*^2/n\right)^{1/\left(2+\beta_Z^{-1}\mathfrak{d}+\beta_X^{-1}\mathfrak{p}\right)} \leq 1$ minimizes $\varepsilon_n^* \asymp \sqrt{\xi \{A + \log(n/\sigma_{\min})\}/n} \vee \delta_{\text{approx}}/\sigma_*$, and gives*

$$\varepsilon_n^* \asymp n^{-\frac{1-\alpha(\beta_Z^{-1}\mathfrak{d}+\beta_X^{-1}\mathfrak{p})}{2+\beta_Z^{-1}\mathfrak{d}+\beta_X^{-1}\mathfrak{p}}} \log^2(n). \tag{14}$$

The proof of Theorem 3 is provided in Appendix J. In the special case when $\alpha = 0$ and $\mathfrak{d} = D$, our convergence rate in (14) recovers the minimax optimal rate for conditional density estimation based on kernel smoothing, as established in (Li et al., 2022).

### 2.4.3. CONDITIONAL DISTRIBUTION ON MANIFOLDS

In this part, we extend Lemma 2 and provide the existence of the generator when the conditional distribution is supported on a compact manifold with dimension $\mathrm{d}_* \leq D$. Due to space constraints, we provide only a sketched proof here; the detailed proof can be found in Appendix D. Specifically, we first present arguments for the existence of the generator when $\mathcal{Y}$ is covered by a single chart. We then extend this to the multiple chart case using the technique of partition of unity.

In the simpler case when there exists a single $(\mathcal{Y},\varphi)$ covering $\mathcal{Y}$, where $\varphi : \mathcal{B}_1(0_{\mathrm{d}_*}) \to \mathcal{Y}$ is a homeomorphism, we assume $\varphi \in \mathcal{H}^{\beta_{\min}+1}$. In this case, we use the change of variable formula to transfer the measure on $\mathcal{B}_1(0_{\mathrm{d}_*})$ (unit ball in $\mathbb{R}^{\mathrm{d}_*}$) from $\mathcal{Y}$. Following Lemma 2, we can find a transport map $g \in \mathcal{H}^{\beta_{\min}}$ mapping from $\mathcal{Z} \times \mathcal{X}$ to $\mathcal{B}_1(0_{\mathrm{d}_*})$. The map $g \circ \varphi$ then serves as our generator.

In the general case where the compact manifold $\mathcal{Y}$ needs to be covered by multiple charts, demonstrating the existence of a transport or push-forward map is challenging because $\mathcal{Y}$ is not uniformly convex. Suppose that $\{(U_k,\varphi_k)\}_{k=1}^K$ forms a cover of $\mathcal{Y}$. Due to the compactness of $\mathcal{Y}$, the

number of charts $K$ is finite. Analogous to the single chart scenario, we first construct $g_k \circ \varphi_k$ to transport the measure on each chart. We then patch these local transport maps together to construct a global transport map; see Appendix D for full details. As a result, following Corollary 1, the convergence rate for density estimation shall be given by $\varepsilon_n^* \asymp n^{-(\beta_{\min}-\mathfrak{d}\alpha)/(2\beta_{\min}+\mathfrak{d})}$.

## 3. Numerical Results

In this section, we present numerical experiments to validate and complement our theoretical findings using two synthetic dataset examples. These experiments cover a range of scenarios, including full-dimensional cases as well as benchmark examples involving manifold-based data. Additionally, we provide a real data example to further enrich our experimentation and validation process. It is worth noting that, although not significant, the computational cost of fitting a conditional generative model is higher compared to fitting an unconditional one, as the input dimension of the deep neural network (DNN) is $\mathfrak{p} + \mathfrak{d}$ rather than just $\mathfrak{d}$.

**Learning algorithm to compute sieve MLE.** For the computational algorithm, we adopt a common conditional variational auto-encoder (VAE) architecture to maximize the following log-likelihood term:$\sum_{j=1}^n \mathcal{L}_{\mathrm{VAE}}(g,\sigma,\phi;Y_j,X_j)$, where

$$\mathcal{L}_{\mathrm{VAE}}(g,\sigma,\phi;y,x) = \log\left(\frac{p_{g,\sigma}(y,x,z)}{q_\phi(Z|y,x)}\right).$$

The variational distribution $q_\phi(Z|y,x)$ is chosen as the standard normal family $\mathsf{N}(\mu_\phi(y,x),\Sigma_\phi(y,x))$.

We examine two classes of datasets: (i) full-dimensional response and (ii) response residing on a low-dimensional manifold. The first highlights the generality of our proposed approach, while the second underscores its efficiency in terms of the Wasserstein metric and validates the small noise perturbation strategy outlined in Corollary 2.

**Simulation from full dimension distribution**. We use the following models for data generation.

- **FD1** : $Y = \mathbb{I}_{\{U<0.5\}}\, \mathsf{N}\left(-X,0.25^2\right) + \mathbb{I}_{\{U>0.5\}}\, \mathsf{N}\left(X,0.25^2\right); U \sim \text{Unif}(0,1), X \sim \mathsf{N}(3,1).$
- **FD2** : $Y = X_1^2 + e^{(X_2+X_3/3)} + \sin(X_4+X_5) + \varepsilon;$ $\{X_j\}_{j=1}^5 \overset{i.i.d}{\sim} \mathsf{N}(0,1), \varepsilon \sim \mathsf{N}(0,1).$
- **FD3** : $Y = X_1^2 + e^{(X_2+X_3/3)} + X_4 - X_5 + 0.5\left(1 + X_2^2 + X_5^2\right) \times \varepsilon; \{X_j\}_{j=1}^5 \overset{i.i.d}{\sim} \mathsf{N}(0,1), \varepsilon \sim \mathsf{N}(0,1).$

These are examples of a mixture model, an additive noise model, and a multiplicative noise model, respectively. The

neural architecture for both the encoder and decoder consists of two deep layers, i.e., $L = 2$. The hyperparameters are as follows: $r_{\text{enc}} = (\mathfrak{p} + 1, 10, 10)$ for $\mu_\phi$ and $\Sigma_\phi$, and $r_{\text{dec}} = (10 + \mathfrak{p}, 10, 1)$ for $g$. The sample size used for simulation is 5000, with a training-to-testing ratio of $4 : 1$. We employ a batch size of 64 with a learning rate of $10^{-3}$.

We compare the sieve MLE with CKDE (Hall et al., 2004) and FlexCode proposed by Izbicki & Lee (2017). To evaluate their performance, we compute the mean squared error (MSE) for both the mean and the standard deviation. We use Monte Carlo approximation to compute the mean and standard deviation for the sieve MLE, and numerical integration for CKDE and Flexcode. This evaluation strategy resembles that implemented by Zhou et al. (2022). Table 2 summarizes the findings.

Table 1. MSE for the estimated conditional mean and the standard deviation.

|     |      | Sieve MLE | CKDE | FlexCode |
| --- | --- | --- | --- | --- |
| FD1 | MEAN | **0.0379** $\pm$ 0.0170 | 1.0053 $\pm$ 0.1004 | 1.1660 $\pm$ 0.1076 |
|     | SD   | **0.0280** $\pm$ 0.0045 | 0.9887 $\pm$ 0.0347 | 1.2000 $\pm$ 0.0126 |
| FD2 | MEAN | **0.1943** $\pm$ 0.0427 | 0.2640 $\pm$ 0.0515 | 0.3954 $\pm$ 0.0571 |
|     | SD   | **0.2843** $\pm$ 0.0093 | 0.2853 $\pm$ 0.0213 | 5.8278 $\pm$ 0.1607 |
| FD3 | MEAN | **0.2337** $\pm$ 0.0453 | 0.2967 $\pm$ 0.0537 | 1.3419 $\pm$ 0.1087 |
|     | SD   | 1.6394 $\pm$ 0.0861 | **0.6334** $\pm$ 0.0460 | 11.4898 $\pm$ 0.1559 |

Note that the sieve MLE outperforms all other methods in all scenarios except for the MSE(SD) for the FD3 dataset. However, for the FD3 dataset, we found that as the training sample size increases further, the MSE(SD) of the sieve MLE achieves performance increasingly comparable to CKDE.

**Simulation from distributions on manifolds.** We consider two examples of manifolds with an intrinsic dimension $\mathfrak{d} = 1$, while the ambient dimension is $D = 2$.

- **M1** : $Y = G_*(Z, U) + \varepsilon$, $G_* = (G_*^{(1)}, G_*^{(2)})$, $G_*^{(1)} = \mathbb{I}_{\{U < 0.5\}}(1 - \cos(Z)) + \mathbb{I}_{\{U > 0.5\}}\cos(Z)$, $G_*^{(2)} = \mathbb{I}_{\{U < 0.5\}}(0.5 - \sin(Z)) + \mathbb{I}_{\{U > 0.5\}}\sin(Z); Z \sim \text{Unif}(0, \pi), U \sim \text{Unif}(0, 1)$.

- **M2** : $Y = G_*(Z, U) + \varepsilon$, $G_* = \left(G_*^{(1)}, G_*^{(2)}\right)$, $G_*^{(1)} = \mathbb{I}_{\{U < 0.5\}}\cos(Z) + \mathbb{I}_{\{U > 0.5\}}2\cos(Z)$, $G_*^{(2)} = \mathbb{I}_{\{U < 0.5\}}0.5\sin(Z) + \mathbb{I}_{\{U > 0.5\}}\sin(Z); Z \sim \text{Unif}(0, 2\pi), U \sim \text{Unif}(0, 1)$.

The manifold $M_1$ consists of two moons. The manifold $M_2$ comprises ellipses, with conditions distinguishing the inner and outer confocal ellipses. The noise sequence follows a two-dimensional centered Gaussian distribution, $\varepsilon \sim \mathsf{N}(0_2, \sigma_*^2 I_2)$. We investigated this setup across various noise variances $\sigma_*^2$. Our neural architecture employed $r_{\text{enc}} = (\mathfrak{p} + 2, 100, 100, 2)$ for $\mu_\phi$ and $\Sigma_\phi$, and

$r_{\text{dec}} = (2 + \mathfrak{p}, 100, 100, 2)$ for $g$. We utilized a sample size of 5000 for simulation, with a training-to-testing ratio of $4 : 1$. A batch size of 100 was employed, with a learning rate of $10^{-3}$.

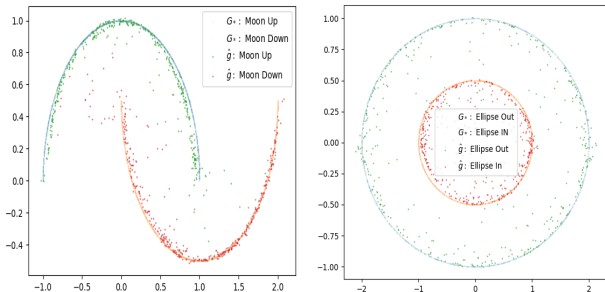

Figure 1. Generated samples from manifold $M_1$ and $M_2$ are displayed.

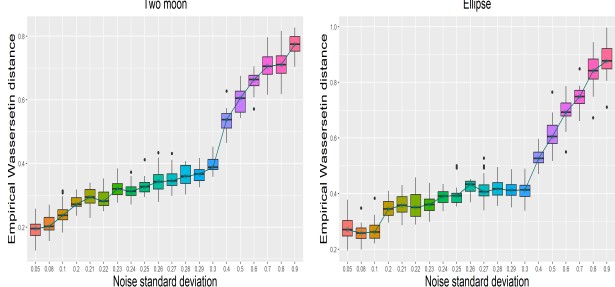

Figure 2. Box plots for the empirical Wasserstein distance at different noise levels $\sigma_*$.

We computed the empirical $W_1$ distance using the algorithm proposed by Cuturi (2013) to evaluate the performance. Figure 2 presents the boxplots of $W_1$ between the true and learned distribution for $M_1$ and $M_2$ across 20 repetitions. The left panel highlights the following general behaviors:

- When $\alpha$ is small and close to zero, the noise variance is large, making estimation challenging due to the singularity of the true data distribution.

- When $\alpha$ is large, the noise variance is small, and the perturbed data facilitates efficient estimation.

This observed pattern, as emphasized in Corollary 2, closely aligns with the results achieved in (13). An additional numerical experiment on real data has been performed and can be found in Appendix A.1.

## 4. Discussion

We investigated statistical properties of a likelihood-based conditional deep generative model for distribution regression in a scenario where the response variable is situated in

a high-dimensional ambient space but is centered around a potentially lower-dimensional intrinsic structure. Our analysis established favorable rates in both the Hellinger and Wasserstein metrics which are dependent on only the intrinsic dimension of the data. Our theoretical findings show that the conditional deep generative models can circumvent the curse of dimensionality for high-dimensional distribution regression. To the best of our knowledge, our work is the first of its kind.

Given the novelty of emerging statistical methodologies with intricate structural considerations in the study of deep generative models, there exist numerous paths for future exploration. Among these potential directions, we are particularly interested in investigating controllable generation via penalized optimization methods, studying statistical properties of deep generative models trained via matching flows, as well as delving into the hypothesis testing problem within the framework of deep generative models, among others. Another interesting direction is to explore residual neural network structure for modeling time series of distributions with interesting temporal dependence structures.

## Impact Statement

This paper presents work whose goal is to advance the field of machine learning theory by understanding the statistical foundations of deep neural network models. There are many potential societal consequences of our work, none which we feel must be specifically highlighted here.

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

# Supplementary Materials for "A Likelihood Based Approach to Distribution Regression Using Conditional Deep Generative Models"

## A. Additional numerical results

### A.1. Numerical result for real data

We utilized the widely used MNIST dataset for two purposes: to demonstrate the generalizability of our approach to a benchmark image dataset where the intrinsic dimension $\mathfrak{d}$ is much lesser than the ambient dimension $D = 784$ and to underscore the effectiveness of sparse networks as outlined in Lemma 4.1 and Corollary 1.1.

For the fully connected architecture, we set $r_{\text{enc}} = (10+784, 512, 2)$ for $\mu_\phi$ and $\Sigma_\phi$, and $r_{\text{dec}} = (10+2, 512, 784)$ for $g$. For the sparse architecture, we use $r_{\text{enc}} = (10 + 784, 608, 432, 256, 2)$ for $\mu_\phi$ and $\Sigma_\phi$, and $r_{\text{dec}} = (10 + 2, 256, 432, 608, 784)$ for $g$. The input dimension of 10 for both the encoder and decoder corresponds to the one-hot encoding of the labels. We employ a batch size of 64 with a learning rate of $10^{-3}$.

Figure 3 presents a visual comparison between real and generated images, organized according to their respective labels. The real images were randomly sampled from the training set along with their corresponding labels, while the generated images were produced using these labels (conditions) and random seeds.

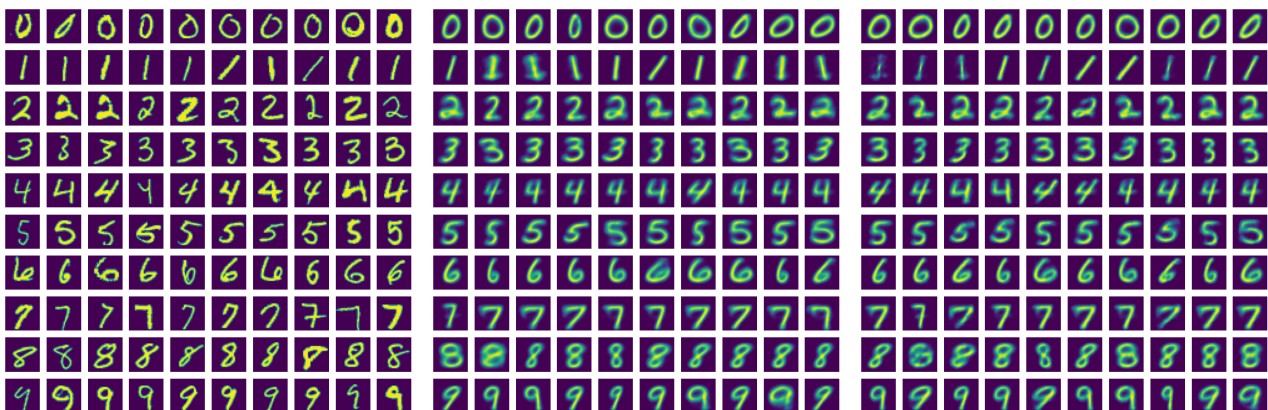

*Figure 3.* MNIST images: real images (left panel), generated images with sparse architecture (central panel), and generated images with fully connected architecture (right panel)

This MNIST example highlights a case where the intrinsic dimension is significantly smaller than the ambient data dimension. This example serves to validate the proposed methodology in high-dimensional settings.

To quantify sample quality, we computed the Wasserstein-1 distance ($W_1$) between generated and test images. For each digit, we averaged $W_1$ distances over 50 samples, reporting results as mean $\pm$ standard deviation. For reference, the baseline $W_1$ distance between two test images is $2.0219 \pm 0.7450$. Table 2 summarizes these distances across different levels of Gaussian noise added during training.

Table 2. Mean W1 distance (± SD) between generated and test MNIST images under varying training-data noise.

| Noise | Sparsely connected | Fully connected |
|---|---|---|
| 0 | $1.9555 \pm 0.7182$ | $1.8859 \pm 0.7355$ |
| 0.005 | $1.9478 \pm 0.7329$ | $1.8663 \pm 0.6251$ |
| 0.01 | $1.9503 \pm 0.7291$ | $1.9598 \pm 0.6867$ |
| 0.02 | $2.0699 \pm 0.6937$ | $2.0616 \pm 0.7410$ |
| 0.04 | $2.2199 \pm 0.6735$ | $2.2117 \pm 0.6627$ |
| 0.06 | $2.3487 \pm 0.6576$ | $2.3172 \pm 0.6267$ |
| 0.08 | $2.4623 \pm 0.6245$ | $2.4076 \pm 0.6308$ |
| 0.1 | $2.5734 \pm 0.6492$ | $2.5002 \pm 0.6337$ |
| 0.3 | $3.4931 \pm 0.7012$ | $3.4943 \pm 0.7164$ |
| 0.5 | $4.0880 \pm 0.7518$ | $4.0995 \pm 0.7633$ |

As shown in Table 2, at zero noise both architectures achieve W1 distances slightly below the baseline, indicating high-fidelity sample generation. As noise increases, W1 distances grow steadily, reflecting degradation in sample quality. Both network types follow similar trends, underlining robustness to architectural choice; minor deviations suggest subtle differences in sensitivity to noise. These empirical observations accord with our theoretical predictions on the large-sample properties of manifold-supported data.

### A.2. Additional numerical results for distributions on manifold

We extended our analysis to examine how the empirical $W_1$ distance varies with sample size, while keeping the noise level fixed at $\sigma_* = 0.01$. Below is a summary table showing the median empirical Wasserstein distances for different sample sizes. The experimental setup remains consistent with the manifold case described in the Section 3.

Table 3. Empirical Wasserstein distance $W_1$ (median) for different sample sizes

| Sample Size | Two Moon ($\sigma_* = 0.01$) | Ellipse ($\sigma_* = 0.01$) |
|---|---|---|
| 4000 | 0.251 | 0.295 |
| 6000 | 0.232 | 0.285 |
| 7000 | 0.216 | 0.271 |
| 8000 | 0.214 | 0.253 |
| 9000 | 0.212 | 0.259 |
| 10000 | 0.196 | 0.251 |

While extracting exact rates through simulation can be challenging, the results in the table validate the large-sample properties for manifolds. These empirical findings align well with the theoretical expectations, further confirming the consistency and convergence trends of our framework.

## B. Notation

We denote $a \vee b$ and $a \wedge b$ as the maximum and minimum of two real numbers $a$ and $b$, respectively. The notation $\lceil a \rceil$ represents the smallest integer greater than or equal to $a$. The inequality $a \lesssim b$ indicates that $a$ is less than or equal to $b$ up to a multiplicative constant. When we write $a \lesssim_{\log} b$, it means that $a$ is less than or equal to $b$ up to a logarithmic factor, specifically $\log(n)$. We denote $a \asymp b$ when both $a \lesssim b$ and $b \lesssim a$ hold. For vector norms, $|\cdot|_{\mathfrak{p}}$ represents the $\ell^{\mathfrak{p}}$ norm, while $\|\cdot\|_{\mathfrak{p}}$ denotes the $L^{\mathfrak{p}}$-norm of a function for $1 \leq \mathfrak{p} \leq \infty$. Lastly, $\mathcal{B}_{\epsilon}(u)$ signifies the Euclidean open ball with radius $\epsilon$ centered at $u$.

We use the multi-index notation through the main paper and the appendix. Denote $\mathbb{N}$ as the set of natural numbers and $\mathbb{N}_0$ as $\mathbb{N} \cup \{0\}$. For a vector $\mathbf{x} \in \mathbb{R}^r$, we denote the components as $\mathbf{x} = (x^{(1)}, \ldots, x^{(r)})$. Given a function $f : D \subset \mathbb{R}^r \to \mathbb{R}$, the operator is defined as $\partial^{\boldsymbol{\alpha}} := \partial^{\alpha^{(1)}} \ldots \partial^{\alpha^{(r)}}$ with $\boldsymbol{\alpha} \in \mathbb{N}_0^r$, where $\partial^{\alpha^{(j)}} f := \partial^{\alpha^{(j)}} f(\mathbf{x})/\partial x^{(j)}$. For $\boldsymbol{\alpha} \in \mathbb{N}_0^r$, the expression $|\boldsymbol{\alpha}| = \sum_{j=1}^r |\alpha^{(j)}|$. Given a function $f(\cdot, \cdot) : D \times D_\prime \subset \mathbb{R}^r \times \mathbb{R}^{r\prime} \to \mathbb{R}$, we denote the operator

$\partial^{\boldsymbol{\alpha}+\boldsymbol{\alpha}_{\prime}} := \partial^{\alpha^{(1)}} \ldots \partial^{\alpha^{(r)}} \partial^{\alpha_{\prime}^{(1)}} \ldots \partial^{\alpha_{\prime}^{(r\prime)}}$, with $\boldsymbol{\alpha} \in \mathbb{N}_0^r$ and $\boldsymbol{\alpha}_{\prime} \in \mathbb{N}_0^{r\prime}$, where $\partial^{\alpha^{(j)}} f(\mathbf{x},\mathbf{y}) = \partial^{\alpha^{(j)}} f(\mathbf{x},\mathbf{y})/\partial^{\alpha^{(j)}} x^{(j)}$ and $\partial^{\alpha_{\prime}^{(j)}} f(\mathbf{x},\mathbf{y}) = \partial^{\alpha_{\prime}^{(j)}} f(\mathbf{x},\mathbf{y})/\partial y^{(j)}$, with $\mathbf{x} \in D$ and $\mathbf{y} \in D_{\prime}$. This notation allows us to represent the derivative with variable $\mathbf{x}$ and $\mathbf{y}$ separately through the vector $\boldsymbol{\alpha}$ and $\boldsymbol{\alpha}_{\prime}$, which is required to tackle the smoothness disparity along $x$ and $y$ variable. The $\beta-$Hölder class functions are defined as

$$\mathcal{H}_r^\beta (D, M) = \Big\{ f : D \subset \mathbb{R}^r \to \mathbb{R} :$$
$$\sum_{\boldsymbol{\alpha}:|\boldsymbol{\alpha}|<\beta} \|\partial^{\boldsymbol{\alpha}} f\|_\infty + \sum_{\boldsymbol{\alpha}:|\boldsymbol{\alpha}|=\lfloor\beta\rfloor} \sup_{\substack{\mathbf{u}_1,\mathbf{u}_2\in D \\ \mathbf{u}_1\neq\mathbf{u}_2}} \frac{|\partial^{\boldsymbol{\alpha}} f(\mathbf{u}_1) - \partial^{\boldsymbol{\alpha}} f(\mathbf{u}_2)|}{|\mathbf{u}_1 - \mathbf{u}_2|_\infty^{\beta-\lfloor\beta\rfloor}} \leq M \Big\}, \tag{15}$$

We extend this definition to include the Hölder class of functions with differences in smoothness (smoothness disparity) along two variables. This class is defined as

$$\mathcal{H}_{r,r_{\prime}}^{\beta,\beta_{\prime}} (D, D_{\prime}, M) = \Big\{ f(\cdot,\cdot) : D \times D_{\prime} \subset \mathbb{R}^r \times \mathbb{R}^{r\prime} \to \mathbb{R} :$$
$$\sum_{\substack{\boldsymbol{\alpha}:|\boldsymbol{\alpha}|<\beta \\ \boldsymbol{\alpha}_{\prime}:|\boldsymbol{\alpha}_{\prime}|<\beta_{\prime}}} \|\partial^{\boldsymbol{\alpha}+\boldsymbol{\alpha}_{\prime}} f\|_\infty + \sum_{\substack{\boldsymbol{\alpha}:|\boldsymbol{\alpha}|=\lfloor\beta\rfloor \\ \boldsymbol{\alpha}_{\prime}:|\boldsymbol{\alpha}_{\prime}|=\lfloor\beta_{\prime}\rfloor}} \sup_{\substack{\mathbf{u}_1,\mathbf{u}_2\in D_X \\ \mathbf{v}_1,\mathbf{v}_2\in D_Y \\ \mathbf{u}_1\neq\mathbf{u}_2 \\ \mathbf{v}_1\neq\mathbf{v}_2}} \frac{|\partial^{\boldsymbol{\alpha}+\boldsymbol{\alpha}_{\prime}} f(\mathbf{v}_1,\mathbf{u}_1) - \partial^{\boldsymbol{\alpha}+\boldsymbol{\alpha}_{\prime}} f(\mathbf{v}_2,\mathbf{u}_2)|}{|\mathbf{u}_1 - \mathbf{u}_2|_\infty^{\beta-\lfloor\beta\rfloor} \vee |\mathbf{v}_1 - \mathbf{v}_2|_\infty^{\beta_{\prime}-\lfloor\beta_{\prime}\rfloor}} \leq M \Big\}. \tag{16}$$

We denote $\mathcal{H}_r^\beta(D) = \cup_{M>0}\mathcal{H}_r^\beta(D,M)$ and $\mathcal{H}_{r,r_{\prime}}^{\beta,\beta_{\prime}}(D,D_{\prime}) = \cup_{M>0}\mathcal{H}_{r,r_{\prime}}^{\beta,\beta_{\prime}}(D,D_{\prime},M)$.

## C. More on Smooth conditional density

**Theorem 4** ((Villani et al., 2009) Theorem 12.50). *Suppose that*

(i) $\mathcal{A}_1$ *and* $\mathcal{A}_2$ *are uniformly convex, bounded, open subsets of* $\mathbb{R}^{\mathfrak{d}}$ *with* $\mathcal{C}^{\lfloor\beta\rfloor+2}$ *(continuously differentiable up to order* $\lfloor\beta\rfloor+2$*) boundaries,*

(ii) $h_1 \in \mathcal{H}^\beta(\mathcal{A}_1)$ *and* $h_2 \in \mathcal{H}^\beta(\mathcal{A}_2)$ *for some* $\beta > 0$*, are probability densities bounded above and below.*

*Then, there exists a unique map (up to an additive constant)* $g : \mathcal{A}_1 \to \mathcal{A}_2$ *with* $g \in \mathcal{H}^{\beta+1}(\mathcal{A}_1)$*, such that if* $U \sim h_1$ *then* $g(U) \sim h_2$*.*

*Proof of Lemma 2.* Given that $Z$ and $X$ is independent, the product measure on $\mathcal{Z} \times \mathcal{X}$ is $p_Z \mu_X^*$. Following the smoothness from $p_Z$ and $\mu_X^*$, the map $p_Z(\cdot)\mu_X^*(\cdot) \in \mathcal{H}^{\min\{\beta_Z,\beta_X\}}(\mathcal{Z} \times \mathcal{X})$. This implies that $p_Z(\cdot)\mu_X^*(\cdot) \in \mathcal{H}^{\min\{\beta_Z,\beta_X,\beta_Q\}}(\mathcal{Z} \times \mathcal{X})$. Again $q_* \in \mathcal{H}^{\beta_Q}(\mathcal{Y})$ implies $q_* \in \mathcal{H}^{\min\{\beta_Z,\beta_X,\beta_Q\}}(\mathcal{Y})$. The result now follows directly from Theorem 4. $\square$

Many of the problems in the conditional setting have an analog in the joint setup. Our proposed approach has a direct statistical extension to this setup. The sufficiency of such extension follows from the observation in the subsequent Lemma 3 which is based on Lemma 2.1 and Lemma 2.2 of Zhou et al. (2022) (see also Theorem 5.10 of Kallenberg (1997)).

**Lemma 3** (Noise Outsourcing Lemma). *Let* $(Y,X) \in \mathcal{Y} \times \mathcal{X}$ *with joint distribution* $P_{Y,X}$*. Suppose* $Y$ *is standard Borel space, then there exists* $Z \sim \mathsf{N}(0, I_m)$ *for any given* $m \geq 1$*, independent of* $X$*, and a Borel measurable function* $G : \mathbb{R}^m \times \mathcal{X} \to \mathcal{Y}$ *such that*

$$(X, G(Z,X)) \sim (Y,X). \tag{17}$$

*Moreover, the condition* (17) *is equivalent of*

$$G(Z,x) \sim P_{Y|X=x}.$$

## D. More on Conditional distribution on manifolds

Suppose $(\mathcal{Y}, \varphi)$ is the single chart covering $\mathcal{Y}$, where $\varphi : \mathcal{B}_1(0_{d_*}) \to \mathcal{Y}$ is a homeomorphism. We assume that $\varphi \in \mathcal{H}^{\beta_{\min}+1}$, and that $\inf_{\mathbf{u}\in\mathcal{B}_1(0_{d_*})} |J_\varphi(\mathbf{u})|$ is bounded below by a positive constant, where

$$|J_\varphi(\mathbf{u})| = \sqrt{\det\left(\frac{\partial\varphi}{\partial\mathbf{u}^\top} \frac{\partial\varphi}{\partial\mathbf{u}}\right)}$$

is the Jacobian determinant of $\varphi$.

Note that when $\mathsf{d}_* < D$, the distribution $Q_*$ cannot possess a Lebesgue density because of the singularity of $\mathcal{Y}$. We, therefore consider a density with respect to the $\mathsf{d}_*$−dimensional Hausdorff measure in $\mathbb{R}^D$, denoted by $\mathsf{H}_{\mathsf{d}_*}$. Suppose that $Q$ allows the Radon-Nikodym derivative $q$ with respect to $\mathsf{H}_{\mathsf{d}_*}$. We further assume that $q$ is bounded from above and below and that $q \circ \varphi \in \mathcal{H}^{\beta_{\min}}$. Then by change of variable formula, the Lebesgue density of $\widetilde{Q}$, the push-forward measure on $\mathcal{B}_1(0_{\mathsf{d}_*})$ through the map $\varphi^{-1}$, is given as

$$\widetilde{q}(\mathbf{u}) = q(\varphi(\mathbf{u}))|J_\varphi(\mathbf{u})|.$$

Following the assumptions on the Jacobian determinant and $\varphi \in \mathcal{H}^{\beta_{\min}+1}$, it follows that $|J_\varphi(\mathbf{u})|$ is bounded from above and below, and the map $\mathbf{u} \mapsto |J_\varphi(\mathbf{u})|$ belongs to $\mathcal{H}^{\beta_{\min}}$. Therefore, $\widetilde{q}$ is bounded above and below, belongs to $\mathcal{H}^{\beta_{\min}}(\mathcal{B}_1(0_{\mathsf{d}_*}))$. By Lemma 2, assuming $\beta_{\min} \le \beta_Z \wedge \beta_X$, there exists $g \in \mathcal{H}^{\beta_{\min}+1}$ such that $\widetilde{Q} = Q_g$. Thus, we have $Q = Q_{\varphi \circ g}$, where $\varphi \circ g : \mathcal{Z} \times \mathcal{X} \to \mathcal{Y}$. Following Lemma 4, it is possible to find the appropriate neural network approximating them.

Suppose $\mathcal{Y}$ is covered by the charts $\{(U_k, \varphi_k)\}_{k=1}^K$, with $1 < K < \infty$, where $\varphi_k : \mathcal{B}_1(0_{\mathsf{d}_*}) \to U_k$ is a homeomorphism. As before, we assume $\varphi_k \in \mathcal{H}^{\beta_{\min}+1}$, $|J_{\varphi_k}(\mathbf{u})|$ is bounded below by a positive constant, $Q$ possesses density $q$ with respect to $\mathsf{H}_{\mathsf{d}_*}$ that is bounded above and below, and that $q \circ \phi_k \in \mathcal{H}^{\beta_{\min}}$. Let $Q_k(\cdot) = Q(\cdot)/Q(U_k)$ be the normalized measure of $Q$ over $U_k$.

We denote $q_k$ as the corresponding density with respect to $\mathsf{H}_{\mathsf{d}_*}$. For $\mathbf{u} \in U_k \cap U_\ell$, $q_k(\mathbf{u})Q(U_k) = q_\ell(\mathbf{u})Q(U_\ell) = q(\mathbf{u})$ holds due to the measure $Q(\cdot)$ being compatible with the charts. This is ensured because the densities $Q(U_k)q_k(\cdot)$ and $Q(U_\ell)q_\ell(\cdot)$ are consistent and align with the measure $Q$ over the overlapping regions of the charts. This compatibility is essential for constructing a coherent global measure from local chart densities.

A compact manifold $\mathcal{Y}$ can be covered by a finite partition of unity $\{\tau_k, k = 1, \ldots, K\}$, each sufficiently smooth (Lee, 2012). By definition, each function in this partition satisfies $\tau_k(\mathbf{u}) = 0$ for $\mathbf{u} \notin U_k$ and $\sum_{k=1}^K \tau_k(\mathbf{u}) = 1$ for all $\mathbf{u} \in \mathcal{Y}$. Given that $q(\mathbf{u}) = Q(U_k)q_k(\mathbf{u})$ for each $k$ and $\mathbf{u} \in U_k$, we can express $q(\mathbf{u})$ as:

$$q(\mathbf{u}) = \sum_{k=1}^K Q(U_k)\tau_k(\mathbf{u})q_k(\mathbf{u}).$$

To normalize, let $c_k = \int \tau_k(\mathbf{u})dQ_k(\mathbf{u})$ and define $q_k'(\mathbf{u}) = \tau_k(\mathbf{u})q_k(\mathbf{u})/c_k$. Thus, we can rewrite $q(\mathbf{u})$ as:

$$q(\mathbf{u}) = \sum_{k=1}^K \pi_k q_k'(\mathbf{u}),$$

where $\pi_k = c_k Q(U_k)$. This formulation reveals that $q$ is a mixture of the component densities $q_k'(\mathbf{u})$, weighted by $\pi_k$. This mixture approach ensures compatibility across different charts, providing a unified density representation over the entire manifold $\mathcal{Y}$.

Since $q_k'$ is sufficiently smooth, we can construct a mapping $g_k : \widetilde{\mathcal{V}} \to \mathcal{Y}$ such that $Q_k'$ is the distribution of $g_k(\widetilde{V})$, supported on $U_k$, where $\widetilde{\mathcal{V}}$ is a uniformly convex set in $\mathbb{R}^{d_*}$, and $\widetilde{V}$ follows a uniform distribution on $\widetilde{\mathcal{V}}$. Next, construct a disjoint partition of the interval $(0, 1)$ into $K$ intervals $I_1, \ldots, I_K$ with lengths $\pi_1, \ldots, \pi_K$, where $I_k = [\sum_{i=1}^{k-1} \pi_i, \sum_{i=1}^k \pi_i]$. Define $h_k$ as the indicator function on the interval $I_k$, i.e., $h_k(u) = 1$ if $u \in I_k$ and 0 otherwise. For a random variable $\mathsf{U}$ following Uniform$(0, 1)$, it follows that $P_\mathsf{U}(h_k(\mathsf{U}) = 1) = \pi_k$, and $P_\mathsf{U}(h_k(\mathsf{U}) = 0) = 1 - \pi_k$. Now, define $\mathbf{v} = (\mathsf{u}, \widetilde{v})$, where $\mathsf{u} \sim$ Uniform$(0, 1)$ and $v \sim$ Uniform$(\widetilde{\mathcal{V}})$. Using this, construct $g(\mathbf{v}) = \sum_{k=1}^K h_k(\mathsf{u})g_k(v)$. It is straightforward to observe that $Q = Q_g$, as the partitioning through $h_k$ ensures that the measure is correctly matched to each $g_k$, and $g_k$ ensures that the restricted distributions $Q_k'$ are appropriately supported on $U_k$.

From an approximation perspective, the indicator functions $h_k$ and the localized generators can be effectively approximated using ReLU neural networks. This also holds for their products and further linear combinations. For details on such constructions, one may refer to Schmidt-Hieber (2019) for sparse neural networks and Kohler et al. (2023) for dense neural networks.

It is important to note that we do not guarantee the regularity of the $g_k$ maps, as they are not necessarily lower bounded. However, the partition of unity maps $\tau_k$ vanish only at the boundary of $U_k$. This property may allows for the construction of sufficiently smooth maps. For the multiple-chart case, we rely on more stringent results, such as Brenier's Theorem (see, for example, Villani et al. (2009)) or the Noise Outsourcing Lemma (Lemma 3), to ensure the existence of the transport maps.

## E. Proof of Lemma 1

*Proof.* For $g_1(\cdot|x), g_2(\cdot|x) \in \mathcal{F}$ with $\||g_1 - g_2|_\infty\|_\infty \leq \eta_1$. Then

$$
\begin{aligned}
&p_{g_1,\sigma}(y|x) - p_{g_2,\sigma}(y|x) \\
&= \int \phi_\sigma(y - g_1(x,z)) \left(1 - \frac{\phi_\sigma(y - g_2(x,z))}{\phi_\sigma(y - g_1(x,z))}\right) dP_Z(z) \\
&= \int \phi_\sigma(y - g_1(x,z)) \left(1 - \exp\left\{-\frac{|y - g_2(x,z)|_2^2 - |y - g_1(x,z)|_2^2}{2\sigma^2}\right\}\right) dP_Z(z) \\
&\leq \int \phi_\sigma(y - g_1(x,z)) \left(\frac{|y - g_2(x,z)|_2^2 - |y - g_1(x,z)|_2^2}{2\sigma^2}\right) dP_Z(z) \quad (18) \\
&= \int \phi_\sigma(y - g_1(x,z)) \left(\frac{|g_2(x,z) - g_1(x,z)|_2^2 - 2(y - g_1(x,z))^T(g_2(x,z) - g_1(x,z))}{2\sigma^2}\right) dP_Z(z) \\
&\leq \int \phi_\sigma(y - g_1(x,z)) \left(\frac{|g_2(x,z) - g_1(x,z)|_2^2}{2\sigma^2} + \frac{2|y - g_1(x,z)|_1|g_2(x,z) - g_1(x,z)|_\infty}{2\sigma^2}\right) dP_Z(z) \\
&\leq \int \phi_\sigma(y - g_1(x,z)) \frac{2KD\eta_1}{2\sigma^2} dP_Z(z) + \frac{2\eta_1}{2\sigma^2} \int |y - g_1(x,z)|_1 \phi_\sigma(y - g_1(x,z)) dP_Z(z) \quad (19) \\
&\leq \frac{2KD\eta_1}{2\sigma^2} \frac{1}{\left(\sqrt{2\pi\sigma^2}\right)^D} + \frac{\eta_1}{\sigma^2} \int \sqrt{\frac{D}{2\pi e}} \frac{1}{(\sqrt{2\pi\sigma^2})^{D-1}} dP_Z(z) \quad (20) \\
&\leq c_1(K,D) \sigma_{\min}^{-(D+2)} \eta_1. \quad (21)
\end{aligned}
$$

For the last line, we use the fact that $\sigma_{\min} \leq 1$. The inequality at (18) follows from $e^{-x} \geq (1-x)$. The ones at (19) follows using

$$
\begin{aligned}
|g_2(x,z) - g_1(x,z)|_2^2 &\leq 2K|g_2(x,z) - g_1(x,z)|_1 \leq 2KD|g_2(x,z) - g_1(x,z)|_\infty \\
&\leq 2KD\||g_1 - g_2|_\infty\|_\infty \leq 2KD\eta_1
\end{aligned}
$$

and $|g_2(x,z) - g_1(x,z)|_\infty \leq \eta_1$. The change at (20) follows from $\phi_\sigma(y - g_1(x,z)) \leq \left(\sqrt{2\pi\sigma^2}\right)^{-D}$ and the bound

$$
|v|_1 \phi_\sigma(v) \leq \sqrt{\frac{D}{2\pi e}} \frac{1}{(\sqrt{2\pi\sigma^2})^{D-1}}.
$$

Now for $\sigma_1, \sigma_2 \in [\sigma_{\min}, \sigma_{\max}]$ with $|\sigma_1 - \sigma_2| \leq \eta_2$. It holds that $\left|\sigma_1^{-2} - \sigma_2^{-2}\right| \leq \sigma_1^{-2}\sigma_2^{-2}(\sigma_1 + \sigma_2)\eta_2$ and $\left|\log\left(\frac{\sigma_2}{\sigma_1}\right)\right| \leq \frac{\eta_2}{\min\{\sigma_1,\sigma_2\}}$. We have

$$
\begin{aligned}
&p_{g,\sigma_1}(y|x) - p_{g_2,\sigma_2}(y|x) \\
&= \int \phi_{\sigma_1}(y - g(x,z)) \left(1 - \left(\frac{\sigma_1}{\sigma_2}\right)^D \exp\left\{\frac{|y - g(x,z)|_2^2}{2}\left(\frac{1}{\sigma_1^2} - \frac{1}{\sigma_2^2}\right)\right\}\right) dP_Z(z) \\
&\leq \int \phi_{\sigma_1}(y - g(x,z)) \left[\frac{|y - g(x,z)|_2^2}{2}\left(\frac{1}{\sigma_2^2} - \frac{1}{\sigma_1^2}\right) - D\log\left(\frac{\sigma_1}{\sigma_2}\right)\right] dP_Z(z) \quad (22) \\
&\leq \int \phi_{\sigma_1}(y - g(x,z)) \left[\frac{|y - g(x,z)|_2^2}{2}\left(\frac{\sigma_1 + \sigma_2}{\sigma_1^2\sigma_2^2}\right)\eta_2 + \frac{D\eta_2}{\min\{\sigma_1,\sigma_2\}}\right] dP_Z(z) \\
&\leq \frac{1}{(\sqrt{2\pi\sigma_1^2})^D} \frac{\sigma_1 + \sigma_2}{e\sigma_2^2}\eta_2 + \frac{1}{\left(\sqrt{2\pi\sigma_1^2}\right)^D} \frac{D\eta_2}{\min\{\sigma_1,\sigma_2\}} \quad (23) \\
&\leq c_2(D)\sigma_{\min}^{-(D+1)}\eta_2. \quad (24)
\end{aligned}
$$

The (22) follows from $1 - e^{-\alpha} \le \alpha$. The change at (23) follows from $\phi_{\sigma_1}(y - g(x,z)) \le \left(\sqrt{2\pi\sigma_1^2}\right)^{-D}$ and

$$|v|_2^2 \phi_\sigma(v) \le \frac{\sigma^2}{(\sqrt{2\pi\sigma^2})^D} \frac{2}{e}.$$

Let $\varepsilon > 0$. Let $\{g_1, \ldots, g_{N_1}\}$ be $\eta_1-$covering of $\mathcal{F}$ and $\{\sigma_1, \ldots, \sigma_{N_2}\}$ be $\eta_2-$covering of $[\sigma_{\min}, \sigma_{\max}]$ with respect to $\||\cdot|_\infty\|_\infty$ and $|\cdot|_\infty$. By (21) and (24), $\eta_1 = c_1^{-1}\sigma_{\min}^{D+2}\varepsilon/4$ and $\eta_2 = c_2^{-2}\sigma_{\min}^{D+1}\varepsilon/4$ implies

$$\left\{P_{g_i, \sigma_j}(\cdot|\cdot) : i = 1, \ldots, N_1, j = 1, \ldots, N_2\right\}$$

forms an $\varepsilon/2-$covering for $\mathcal{P}$ with respect to $\|\cdot\|_\infty$. Denote the envelope function of $\mathcal{F}$

$$H(y, x) = \sup_{p \in \mathcal{P}} p(y|x) \le \frac{1}{(2\pi\sigma_{\min}^2)^{-D/2}} \exp\left\{-\frac{|y|_2^2 - 4K^2 D}{4\sigma_{\max}^2}\right\}$$

$$= e^{K^2 D/2\sigma_{\max}^2} 2^{D/2} \left(\frac{\sigma_{\max}}{\sigma_{\min}}\right)^D \phi_{\sqrt{2}\sigma_{\max}}(y).$$

Following from $\int_{|y|_\infty > t} \phi_\sigma(y)dy \le 2D e^{-t^2/2\sigma^2}$, we have

$$\int\int_{|y|_\infty > B} H(y, x)\mu(y, x)dydx = \int\left(\int_{|y|_\infty > B} H(y, x)\mu(y|x)dy\right)\mu_X^*(x)dx < \varepsilon,$$

where

$$B = 2\sigma_{\max}\left(\log\frac{1}{\varepsilon} + D\log\frac{\sigma_{\max}}{\sigma_{\min}} + \frac{K^2 D}{2\sigma_{\max}^2} + \log 2D\right)^{1/2}.$$

For each $(i, j)$ define

$$l_{ij}(y, x) = \max\left\{p_{g_i, \sigma_j}(y, x) - \varepsilon/2, 0\right\} \quad \text{and} \quad u_{ij}(y, x) = \min\left\{p_{g_i, \sigma_j}(y, x) + \varepsilon/2, H(y, x)\right\}.$$

It follows that

$$\int\int\left\{u_{ij}(y, x) - l_{ij}(y, x)\right\}\mu_X^*(x)dydx$$
$$\le \int\int_{|y|_\infty \le B} \varepsilon\mu_X^*(x)dydx + \int\int_{|y|_\infty > B} H(y, x)\mu_X^*(x)dydx \tag{25}$$
$$\le \left\{(2B)^D + 1\right\}\varepsilon.$$

Denote $\delta^2 := \left\{(2B)^D + 1\right\}$. With $d_H^2(u_{ij}, l_{ij}) \le d_1(u_{ij}, l_{ij})$, we have

$$\mathcal{N}_{[]}(\delta, \mathcal{P}, d_H) \le \mathcal{N}_{[]}(\delta^2, \mathcal{P}, d_1) \le N_1 N_2 \le \frac{\sigma_{\max} - \sigma_{\min}}{\eta_2}\mathcal{N}(\eta_1, \mathcal{F}, \||\cdot|_\infty\|_\infty). \tag{26}$$

It is possible to write

$$\delta^2 = \varepsilon \le C_1(\sigma_{\max}, D)\left[\varepsilon(\log\varepsilon^{-1})^{D/2} + \varepsilon C_2(K) + \varepsilon\left(\log\frac{\sigma_{\max}}{\sigma_{\min}}\right)^{D/2}\right],$$

where $C_1(\sigma_{\max}, D)$ and $C_2(K)$ is a constant. There exists small enough $\varepsilon_*(D)$ such that for all $\varepsilon \in (0, \varepsilon_*]$

$$\delta^2 \le C_3(\sigma_{\max}, D, K)\sqrt{\varepsilon}\left(\log\frac{\sigma_{\max}}{\sigma_{\min}}\right)^{D/2}.$$

Consequently, there exists $\delta_* = \delta_*(D)$, such that for all $\delta \le \delta_*$, we have

$$C_3^2(\sigma_{\max}, K, D)\delta^4\left(\log\frac{\sigma_{\max}}{\sigma_{\min}}\right)^{-D} \le \varepsilon.$$

It lead us to, for all $\delta \leq \delta_*$

$$\eta_1 \geq \frac{c_1^{-1} C_3^2 \sigma_{\min}^{D+3} \delta^4}{\sigma_{\min}\{\log(\sigma_{\max}/\sigma_{\min})\}^D} \geq c\sigma_{\min}^{D+3}\delta^4, \tag{27}$$

where $c(\sigma_{\max}, K, D)$ is a constant. We use the fact that $\sigma_{\min}\{\log(\sigma_{\max}/\sigma_{\min})\}^D$ is bounded above by some constant depending only upon $\sigma_{\max}$ as $\sigma_{\min} \leq 1$. Similar to (27), it is possible to write for all $\delta > \delta_*$

$$\eta_2 \geq c'\sigma_{\min}^{D+2}\delta^4, \qquad \text{for all } \delta \leq \delta_*, \tag{28}$$

where $c'(\sigma_{\max}, K, D)$ is some constant.

The result now follows directly (28) and (27) with (26). $\qquad\square$

## F. Proof of Theorem 1

*Proof.* Choose four absolute constants $c_1, \ldots, c_4$ as in Theorem 1 of Wong & Shen (1995). Define $c$ and $C$ in the statement of Lemma 1. The proof closely follows Chae et al. (2023). We have therein the proof of Theorem 3 that

$$\int_{\varepsilon^2/2^8}^{\sqrt{2}\varepsilon} \sqrt{\log \mathcal{N}_{[]}(\delta/c_3, \mathcal{P}, d_H)}d\delta$$
$$\leq \sqrt{2}\varepsilon\sqrt{\xi A + (D+3)(s+1)\log\sigma_{min}^{-1} + c_5\xi} + \sqrt{2}\varepsilon\sqrt{4(\xi+1)}\sqrt{\log(2^8/\varepsilon^2)}, \tag{29}$$

for every $\varepsilon \leq \sqrt{2} \leq c_3\delta_*/\sqrt{2}$, where $c_5 = c_5(c, C, c_3)$. Observe that $c_4\sqrt{n}\varepsilon_n^2$ is upper bound to (29) and Eq. (3.1) of Wong & Shen (1995) is satisfied.

Using B.12 of Ghosal & van der Vaart (2017), we have

$$K(p_{G_*,\sigma_*}, p_{g,\sigma_*}) \leq \int\int K\Big(N\big(G_*(z,x),\sigma_*^2\big), N\big(g(z,x),\sigma_*^2\big)\Big)\mu_X^*(x)\,dx\,dP_Z(z)$$
$$= \int\int \frac{|G_*(z,x) - g(z,x)|_2^2}{2\sigma_*^2}\mu_X^*(x)\,dx\,dP_Z(z) \leq \frac{D\delta_{\text{approx}}^2}{2\sigma_*^2} =: \delta_n.$$

One may easily see that

$$\int\left(\log\frac{\phi_\sigma(x)}{\phi_\sigma(x-y)}\right)^2\phi_\sigma(x)dx = \int\frac{|y|_2^4 + 4|x^T y|^2}{4\sigma^2}\phi_\sigma(x)dx \leq \frac{|y|_2^4}{4\sigma^2} + |y|_2^2\int\frac{|x|_2^2}{\sigma^2}\phi_\sigma(x)dx.$$

Combining this with Example B.12, (B.17) and Exercise B.8 of Ghosal & van der Vaart (2017), we have

$$\int\int\left(\log\frac{p_{G_*,\sigma_*}(y|x)}{p_{g,\sigma_*}(y|x)}\right)^2 dP_*(y|x)\,\mu_X^*(x)dx$$
$$\leq \int\int\int\left(\log\frac{\phi_\sigma(y - G_*(z,x))}{\phi_\sigma(y - G(z,x))}\right)^2\phi_\sigma(y - G_*(z,x))\,dy\,dP_Z(z)\,\mu_X^*(x)dx$$
$$\leq \frac{D^2\delta_{\text{approx}}^4}{4\sigma_*^2} + D\delta_{\text{approx}}^2\int\frac{|x|_2^2}{\sigma_*^2}\phi_{\sigma_*}(y)dy + \frac{2D\delta_{\text{approx}}^2}{\sigma_*^2} \leq c_7\frac{\delta_{\text{approx}}^2}{\sigma_*^2} =: \tau_n,$$

where $c_7 = c_7(D)$. We are using $\delta_n$ and $\tau_n$, although they are independent of $n$, for notational consistency with Theorem 4 of Wong & Shen (1995). Let $\varepsilon_n^* = \varepsilon_n \vee \sqrt{12\delta_n}$. Then, using Theorem 4 of Wong & Shen (1995), we have

$$P_*\left(d_H(\widehat{p}, p_*) > \varepsilon_n\right) \leq 5e^{-c_2 n\varepsilon_n^{*2}} + \frac{\tau_n}{n\delta_n} = 5e^{-c_2 n\varepsilon_n^{*2}} + \frac{2c_7^2}{Dn}.$$

The proof is complete after redefining constants. $\qquad\square$

## G. Proofs of Corollary 1

*Proof.* For the sparse case in 1.1, utilizing the entropy bound from (10), we observe that

$$\xi\{A + \log(n/\sigma_{\min})\} \asymp \delta_{\mathrm{approx}}^{-t_*/\beta_*} \log^3(\delta_{\mathrm{approx}}^{-1}),$$

which naturally leads to the required convergence rate.

Similarly for the fully connected case 1.2, utilizing the entropy bound from (11) , we observe that

$$\xi\{A + \log(n/\sigma_{\min})\} \asymp \delta_{\mathrm{approx}}^{-t_*/\beta_*} \log^3(\delta_{\mathrm{approx}}^{-1}),$$

which naturally leads to the required convergence rate. $\qquad\square$

## H. Proof of Theorem 2

*Proof.* It is suffice to assume that $\varepsilon$ and $\sigma_* \sqrt{\log \varepsilon^{-1}}$ are sufficiently small. If not, let $\varepsilon + \sigma_* \sqrt{\log \varepsilon^{-1}} \geq c_0$, where $c_0(K, D, r_*)$. Then Theorem 2 holds trivially by taking a large enough constant depending just on $D$, $K$, and $r_*$.

Let $V \sim Q(\cdot|X = x)$, $V_* \sim Q(\cdot|X = x)$, $\boldsymbol{\epsilon} \sim \mathsf{N}(0_D, \sigma^2 \mathbb{I}_d)$ and $\boldsymbol{\epsilon}_* \sim \mathsf{N}(0_D, \sigma_*^2 \mathbb{I}_d)$ be independent with underlying probability density $\nu$. We truncate the random variable $\boldsymbol{\epsilon}$ and $\boldsymbol{\epsilon}_*$ componentwise as $(\boldsymbol{\epsilon}_K)_j = \max\{-K, \min\{K, \boldsymbol{\epsilon}_j\}\}$ and $(\boldsymbol{\epsilon}_{*K})_j = \max\{-K, \min\{K, (\boldsymbol{\epsilon}_*)_j\}\}$ respectively. We denote $P_{g,\sigma}$ as $P$, $Q_g$ as $Q$, $\widetilde{P}$ as distribution of $V + \boldsymbol{\epsilon}_K$ and $\widetilde{P}_*$ as the distribution of $V_* + \boldsymbol{\epsilon}_{*K}$. One may note that $W_1(\widetilde{P}_*, Q_*) \leq W_2(\widetilde{P}_*, Q_*) \leq \sqrt{\mathbb{E}\big[|\boldsymbol{\epsilon}_{*K}|_2^2\big]} \leq \sqrt{\mathbb{E}\big[|\boldsymbol{\epsilon}_*|_2^2\big]} \leq \sigma_* \sqrt{D}$. Similarly, $W_1(\widetilde{P}, Q) \leq \sigma \sqrt{D}$. The $\ell_1$ diameter of $[-2K, 2K]^D$, where the support of $\widetilde{P}$ and $\widetilde{P}_*$, is 4KD. Observe that

$$W_1\left(\widetilde{P}_*, \widetilde{P}\right) \leq 4\,KD\,d_1\left(\widetilde{P}_*, \widetilde{P}\right) \leq 4\,KD\,d_1(P_*, P) \leq 8\,KD\,d_H(P_*, P),$$

where the first inequality follows from Theorem 4 of Gibbs & Su (2002), the second inequality follows from the fact the distance between two truncated distributions is always lesser than the original distributions and the last inequality follows from $d_1 \leq 2d_H$. Hence,

$$W_1\left(Q_*, Q\right) \leq W_2\left(Q_*, \widetilde{P}_*\right) + W_1\left(\widetilde{P}_*, \widetilde{P}\right) + W_2\left(\widetilde{P}, Q\right) \leq \sigma_* \sqrt{D} + 8\,KD\,\varepsilon + \sigma \sqrt{D}.$$

Now it is suffice to show that $\sigma \leq c\,\sigma_* \sqrt{\log \varepsilon^{-1}}$, where $c = c(D, K, r*)$ is a constant, because we have assumed that $\varepsilon$ is small enough. We establish this in the rest of the proof. Let $t_* = \left[2\,\sigma_*^2\,D\log\left(\frac{2D}{\varepsilon}\right)\right]^{1/2}$. Observe that

$$\int_{|x|_2 > t_*} \phi_{\sigma_*}(x)dx \leq \int_{|x|_\infty > t_*/\sqrt{D}} \phi_{\sigma_*}(x)dx \leq 2\,D e^{-t_*^2/2D\sigma^2} \leq \varepsilon.$$

Let $\mathcal{M}_*^{t_*} = \mathcal{M}_* \oplus \mathcal{B}_{t_*}(0_D)$. We may write

$$
\begin{aligned}
1 - P_*\left(\mathcal{M}_*^{t_*}\right) &= \nu\left(Y_* + \boldsymbol{\epsilon}_* \notin \mathcal{M}_*^{t_*}\right) \leq \nu\left(|\boldsymbol{\epsilon}_*|_2 > t_*\right) \\
\implies P\left(\mathcal{M}_*^{t_*}\right) &\geq 1 - 2\varepsilon,
\end{aligned}
\tag{30}
$$

the implication in the last line follows from $\sup_B |P(B) - P_*(B)| \leq d_H(P, P_*) \leq \varepsilon$. For the sake of contradiction, let $\sigma \in [2t_*, r^*/2] \cup (r_*/2, \infty)$ ($t_*$ is sufficiently small, from the assumption we made at the beginning of this proof). If $\sigma > r_*/2$, then

$$2\varepsilon \geq 1 - P\left(\mathcal{M}_*^{t_*}\right) \geq 1 - P\left([-K, K]^D\right) \geq c_2(K, D, r*)$$

where $c_2$ is some positive constant. It is a contradiction following from the smallness of $\varepsilon$. Lets make a claim that if $\sigma \in [2t_*, r_*/2]$, then for every $y \in \mathbb{R}^D$, there is some $z \in \mathbb{R}^D$ such that $|z - y|_2 \leq \sigma$ and $\mathcal{B}_{\sigma/2}(z) \cap \mathcal{M}_*^{t_*} = \emptyset$.

Following from the claim, we have

$$\nu\left(Y + \boldsymbol{\epsilon} \notin \mathcal{M}_*^{t_*} \big| Y = y\right) \geq \nu\left(\boldsymbol{\epsilon} \in \mathcal{B}_{\sigma/2}(z - y)\right).$$

Since $|z - y|_2 \leq \sigma$, the right hand side is bounded below by a positive constant depending just on $D$ which is again a contradiction to (30). This proves the assertion made in the theorem.

The proof of the claim is divided into three cases. Let $\rho(y, \mathcal{M}_*) = \inf\{|y - y'|_2 : y' \in \mathcal{M}_*\}$ be the $\ell_2$ set distance.

**Case 1.** $\rho(y, \mathcal{M}_*) \geq \sigma$ : We may choose $z = y$.

**Case 2.** $\rho(y, \mathcal{M}_*) \in (0, \sigma)$ : Let $y_0$ be the unique Euclidean projection of $y$ onto $\mathcal{M}_*$. Such a unique projection exists because $\sigma < r_*$ is within the reach and $y \in \mathcal{M}_*$, since $\mathcal{M}_*$ is closed. Suppose $y_t = y_0 + t(y - y_0)$. We shall define two continuous functions $d_0(t) = |y_t - y_0|_2$ and $d(t) = \rho(y_t, \mathcal{M}_*)$. It is obvious that $d(t) \leq d_0(t)$. For $t \in [0, 1 + \sigma/|y - y_0|_2]$, $d_0(t) \leq d(t)$ because $y_0$ is the unique projection for all the points that lie on the line segment including the farthest point with $t = 1 + \sigma/|y - y_0|_2$. Otherwise, say $d(t) = \rho(y_t, z)$ and

$$|y - y_0|_2 = |y - y_t|_2 + |y_t - y_0|_2 > |y - y_t| + |y_t - z| \geq |y - z|_2$$

which contradicts $y_0$ being a unique projection. The claim holds for the point $z = y_{1+\sigma/|y-y_0|_2}$. To see this, observe $|z - y| = \sigma$ and $\mathcal{B}_{\sigma/2}(z) \cap \mathcal{M}_*^{t_*} = \emptyset$ because $t_* \leq \sigma/2$ and the ball $\mathcal{B}_{\sigma/2}(z) \subset \mathcal{M}_*^{r_*}$ is within the reach of the manifold.

**Case 3.** $\rho(y, \mathcal{M}_*) = 0$ : Because $\mathcal{M}_*$ has empty interior, for all $\gamma > 0$, we always find a point $y_\gamma$, which in $\mathcal{B}_\gamma(y)$ which away from $\mathcal{M}_*$. For small enough $\gamma$, we reduce to case 2 by taking $\gamma \to 0$, the limit point of $y_\gamma$ has the required behavior.

$\square$

# I. Proof of Corollary 2

*Proof.* The effective noise variance after the perturbation would be

$$\widetilde{\sigma}_* = n^{-\alpha} + n^{-\beta_*/2(\beta_* + t_*)} \asymp \begin{cases} n^{-\alpha}, & \alpha < \beta_*/\{2(\beta_* + t_*)\} \\ n^{\beta_*/2(\beta_* + t_*)}, & \text{otherwise.} \end{cases}$$

Following this and the Theorem 2, for the rate we have

$$\varepsilon_n^* + \sigma_* \sqrt{\log((\varepsilon_n^*)^{-1})} \asymp \left( n^{-\frac{\beta_* - t_* \alpha}{2\beta_* + t_*}} + n^{-\alpha} \right) \log^2(n)$$

$$\asymp \begin{cases} n^{-\frac{\beta_* - t_* \alpha}{2\beta_* + t_*}} \log^2(n), & \text{if } \alpha < \beta_*/\{2(\beta_* + t_*)\}, \\ n^{-\frac{\beta_*}{2(\beta_* + t_*)}} \log^2(n), & \text{otherwise.} \end{cases}$$

$\square$

# J. Proof of Theorem 3

*Proof.* With $m = \lceil \log_2(n) \rceil$ and $N = \left( n^{(\beta_Z^{-1}d + \beta_X^{-1}p)[1 + \alpha(\beta_Z^{-1}d + \beta_X^{-1}p)]/[2 + \beta_Z^{-1}d + \beta_X^{-1}p]} \right)$ in Theorem 5, we can find a network $G$ with the mentioned architecture such that

$$\||G - G_*|_\infty\|_\infty \leq \delta_{\text{approx}}.$$

Following the entropy bound from (10), we have

$$\log \mathcal{N}(\delta, \mathcal{F}_s, \|| \cdot |_\infty\|_\infty) \lesssim sL \{\log(rL) + \log \delta^{-1}\}$$

$$\lesssim \delta_{\text{approx}}^{-(\beta_Z^{-1}d + \beta_X^{-1}p)} \log^2 \delta_{\text{approx}}^{-1} \{ \log \left( \delta_{\text{approx}}^{-1} \log \left( \delta_{\text{approx}}^{-1} \right) \right) + \log \left( \delta_{\text{approx}}^{-1} \right) \}.$$

The rest directly follows from the Theorem 1

$\square$

# K. Approximation properties of the sparse and fully connected DNNs

The approximability of the sparse network is detailed in Lemma 4.1, which restates Lemma 5 from Chae et al. (2023). For the fully connected network, Lemma 4.2 demonstrates its approximation capabilities, derived directly from Theorem 2 and

the proof of Theorem 1 in Kohler & Langer (2021). Additionally, the inclusion of the class $\mathcal{G}$ in the fully connected setup is supported by the discussion in Section 1 of Kohler & Langer (2020).

**Lemma 4.** *Suppose that $G_* \in \mathcal{G}$. Then, for every small enough $\delta \in (0,1)$,*

1. *there exists a sparse network $G \in \mathcal{F}_s = \mathcal{F}_s(L, r, s, K \vee 1)$ with $L \lesssim \log \delta^{-1}$, $r \lesssim \delta^{-t_*/\beta_*}$, $s \lesssim \delta^{-t_*/\beta_*} \log \delta^{-1}$ satisfying $\||G - G_*|_\infty\|_\infty \leq \delta$.*

2. *there exists a fully connected network $G \in \mathcal{F}_c$ with $L \lesssim \log \delta^{-1}$, $r \lesssim \delta^{-t_*/2\beta_*}$, $B \lesssim \delta^{-1}$ satisfying $\||G - G_*|_\infty\|_\infty \leq \delta$.*

## L. A new approximation result for functions with smoothness disparity

In this section, we prove the approximability of the sparse neural network for the Hölder class of function $f \in \mathcal{H}_{r,r'}^{\beta,\beta'}(D, D', K)$.

**Theorem 5.** *Let $f \in \mathcal{H}_{r,r'}^{\beta,\beta'}([0,1]^r, [0,1]^{r'}, K)$. Denote $\mathsf{r}_{\mathrm{sum}} = r + r'$ and $\beta_{\mathrm{sum}} = \beta + \beta'$. Then for any integers $m \geq 1$ and $N \geq (\beta_{\mathrm{sum}} + 1)^{\mathsf{r}_{\mathrm{sum}}} \vee (K + 1)e^{\mathsf{r}_{\mathrm{sum}}}$, there exists a network*

$$\widetilde{f} \in \mathcal{F}_s\big(L, \big(\mathsf{r}_{\mathrm{sum}}, 6(\mathsf{r}_{\mathrm{sum}} + \lceil \beta_{\mathrm{sum}} \rceil)N, \ldots, 6(\mathsf{r}_{\mathrm{sum}} + \lceil \beta_{\mathrm{sum}} \rceil)N, 1\big), s, \infty\big)$$

*with depth*

$$L = 8 + (m + 5)\big(1 + \lceil \log_2 (\mathsf{r}_{\mathrm{sum}} \vee \beta_{\mathrm{sum}}) \rceil\big)$$

*and the number of parameters*

$$s \leq 109\big(\mathsf{r}_{\mathrm{sum}} + \beta_{\mathrm{sum}} + 1\big)^{3 + \mathsf{r}_{\mathrm{sum}}} N(m + 6),$$

*such that*

$$\|\widetilde{f} - f\|_{L^\infty([0,1]^{\mathsf{r}_{\mathrm{sum}}})} \leq (2K + 1)\big(1 + \mathsf{r}_{\mathrm{sum}}^2 + \beta_{\mathrm{sum}}^2\big) 6^{\mathsf{r}_{\mathrm{sum}}} N 2^{-m} + K\, 3^{\mathsf{r}_{\mathrm{sum}}/(\beta^{-1}r + \beta'^{-1}r')} N^{-1/(\beta^{-1}r + \beta'^{-1}r')}.$$

We denote $\widetilde{\beta} = (\beta + \beta')^{-1}\beta\beta'$ and $\widetilde{r} = (\beta + \beta')^{-1}(r\beta + r'\beta')$. Before presenting the proof of Theorem 5, we formulate some required results.

We follow the classical idea of function approximation by local Taylor approximations that have previously been used for network approximations in (Yarotsky, 2017) and (Schmidt-Hieber, 2020). For a vector $\mathbf{a} \in [0,1]^r$ define

$$P_{\mathbf{a},\mathbf{b}}^{\beta,\beta'} f(\mathbf{u}, \mathbf{v}) = \sum_{\substack{0 \leq |\boldsymbol{\alpha}| < \beta \\ 0 \leq |\boldsymbol{\alpha}'| < \beta'}} (\partial^{\boldsymbol{\alpha} + \boldsymbol{\alpha}'} f)(\mathbf{a}, \mathbf{b}) \frac{(\mathbf{u} - \mathbf{a})^{\boldsymbol{\alpha}}(\mathbf{v} - \mathbf{b})^{\boldsymbol{\alpha}'}}{\boldsymbol{\alpha}!\, \boldsymbol{\alpha}'!}. \tag{31}$$

We use the notation the $\mathbf{u} = (u^{(j)})_j$ to represent the component of the vector when the index $j$ is well understood. Accordingly we have $\mathbf{v} = (v^{(j)})_j$, $\mathbf{a} = (a^{(j)})_j$ and $\mathbf{b} = (b^{(j)})_j$. By Taylor's theorem for multivariate functions, we have for a suitable $\xi \in [0,1]$,

$$f(\mathbf{u}, \mathbf{v}) = \sum_{\substack{\boldsymbol{\alpha}:|\boldsymbol{\alpha}| < \beta - 1 \\ \boldsymbol{\alpha}':|\boldsymbol{\alpha}'| < \beta' - 1}} (\partial^{\boldsymbol{\alpha} + \boldsymbol{\alpha}'} f)(\mathbf{a}, \mathbf{b}) \frac{(\mathbf{u} - \mathbf{a})^{\boldsymbol{\alpha}}(\mathbf{v} - \mathbf{b})^{\boldsymbol{\alpha}'}}{\boldsymbol{\alpha}!\, \boldsymbol{\alpha}'!}$$
$$+ \sum_{\substack{\beta - 1 \leq |\boldsymbol{\alpha}| < \beta \\ \beta' - 1 \leq |\boldsymbol{\alpha}'| < \beta'}} (\partial^{\boldsymbol{\alpha} + \boldsymbol{\alpha}'} f)(\mathbf{a} + \xi(\mathbf{u} - \mathbf{a}), \mathbf{b} + \xi(\mathbf{v} - \mathbf{b})) \frac{(\mathbf{u} - \mathbf{a})^{\boldsymbol{\alpha}}(\mathbf{v} - \mathbf{b})^{\boldsymbol{\alpha}'}}{\boldsymbol{\alpha}!\, \boldsymbol{\alpha}'!}.$$

We have $|(\mathbf{u} - \mathbf{a})^{\boldsymbol{\alpha}}| = \prod_{j=1}^{r} |u_j - a_j|^{\alpha^{(j)}} \leq |\mathbf{u} - \mathbf{a}|_{\infty}^{|\boldsymbol{\alpha}|}$ and $|(\mathbf{v} - \mathbf{b})^{\boldsymbol{\alpha}_{\prime}}| = \prod_{j=1}^{r_{\prime}} |v_j - b_j|^{\alpha_{\prime}^{(j)}} \leq |\mathbf{v} - \mathbf{b}|_{\infty}^{|\boldsymbol{\alpha}_{\prime}|}$. Consequently, for $f \in \mathcal{H}_{r,r_{\prime}}^{\beta,\beta_{\prime}}([0,1]^r, [0,1]^{r_{\prime}}, K)$,

$$
\begin{aligned}
&\left| f(\mathbf{u}, \mathbf{v}) - P_{\mathbf{a},\mathbf{b}}^{\beta,\beta_{\prime}} f(\mathbf{u}, \mathbf{v}) \right| \\
&\leq \sum_{\substack{\beta-1 \leq |\boldsymbol{\alpha}| < \beta \\ \beta_{\prime}-1 \leq |\boldsymbol{\alpha}_{\prime}| < \beta_{\prime}}} \left( \partial^{\boldsymbol{\alpha}+\boldsymbol{\alpha}_{\prime}} f(\mathbf{a} + \xi(\mathbf{u} - \mathbf{a}), \mathbf{b} + \xi(\mathbf{v} - \mathbf{b})) - \partial^{\boldsymbol{\alpha}+\boldsymbol{\alpha}_{\prime}} f(\mathbf{a}, \mathbf{b}) \right) \frac{(\mathbf{u} - \mathbf{a})^{\boldsymbol{\alpha}} (\mathbf{v} - \mathbf{b})^{\boldsymbol{\alpha}_{\prime}}}{\boldsymbol{\alpha}! \, \boldsymbol{\alpha}_{\prime}!} \\
&\leq K \left( |\mathbf{u} - \mathbf{a}|_{\infty}^{\beta} \vee |\mathbf{v} - \mathbf{b}|_{\infty}^{\beta_{\prime}} \right)
\end{aligned}
\tag{32}
$$

We may also write (31) as a linear combination of monomials

$$
P_{\mathbf{a},\mathbf{b}}^{\beta,\beta_{\prime}} f(\mathbf{u}, \mathbf{v}) = \sum_{\substack{0 \leq |\boldsymbol{\gamma}| < \beta \\ 0 \leq |\boldsymbol{\gamma}_{\prime}| < \beta_{\prime}}} c_{\boldsymbol{\gamma},\boldsymbol{\gamma}_{\prime}} \mathbf{u}^{\boldsymbol{\gamma}} \mathbf{v}^{\boldsymbol{\gamma}_{\prime}},
\tag{33}
$$

for suitable coefficients $c_{\boldsymbol{\gamma},\boldsymbol{\gamma}_{\prime}}$. For convenience, we omit the dependency on $\mathbf{a}$ and $\mathbf{b}$ in $c_{\boldsymbol{\gamma},\boldsymbol{\gamma}_{\prime}}$. Since $\partial^{\boldsymbol{\gamma},\boldsymbol{\gamma}_{\prime}} P_{\mathbf{a},\mathbf{b}}^{\beta,\beta_{\prime}} f(\mathbf{u}, \mathbf{v}) |_{(\mathbf{u}=0, \mathbf{v}=0)} = \boldsymbol{\gamma}! \, \boldsymbol{\gamma}_{\prime}! \, c_{\boldsymbol{\gamma},\boldsymbol{\gamma}_{\prime}}$, we must have

$$
c_{\boldsymbol{\gamma},\boldsymbol{\gamma}_{\prime}} = \sum_{\substack{\boldsymbol{\gamma} \leq \boldsymbol{\alpha} \, \& \, |\boldsymbol{\alpha}| < \beta \\ \boldsymbol{\gamma}_{\prime} \leq \boldsymbol{\alpha}_{\prime} \, \& \, |\boldsymbol{\alpha}_{\prime}| < \beta_{\prime}}} (\partial^{\boldsymbol{\alpha}+\boldsymbol{\alpha}_{\prime}} f)(\mathbf{a}, \mathbf{b}) \frac{(-\mathbf{a})^{\boldsymbol{\alpha}-\boldsymbol{\gamma}} (-\mathbf{b})^{\boldsymbol{\alpha}_{\prime}-\boldsymbol{\gamma}_{\prime}}}{\boldsymbol{\gamma}! \, \boldsymbol{\gamma}_{\prime}! \, (\boldsymbol{\alpha}-\boldsymbol{\gamma})! \, (\boldsymbol{\alpha}_{\prime}-\boldsymbol{\gamma}_{\prime})!}.
$$

Notice that since $\mathbf{a} \in [0,1]^r$, $\mathbf{b} \in [0,1]^{r_{\prime}}$, and $f \in \mathcal{H}_{r,r_{\prime}}^{\beta,\beta_{\prime}}([0,1]^r, [0,1]^{r_{\prime}}, K)$,

$$
|c_{\boldsymbol{\gamma}\boldsymbol{\gamma}_{\prime}}| \leq K/(\boldsymbol{\gamma}! \, \boldsymbol{\gamma}_{\prime}!) \quad \text{and} \quad \sum_{\substack{\boldsymbol{\gamma} \geq 0 \\ \boldsymbol{\gamma}_{\prime} \geq 0}} |c_{\boldsymbol{\gamma},\boldsymbol{\gamma}_{\prime}}| \leq K \prod_{i=1}^{r} \prod_{j=1}^{r_{\prime}} \sum_{\gamma^{(i)} \geq 0} \sum_{\gamma_{\prime}^{(j)} \geq 0} \frac{1}{\gamma^{(i)}!} \frac{1}{\gamma_{\prime}^{(j)}!} = K e^{r+r_{\prime}},
\tag{34}
$$

where $\boldsymbol{\gamma} = (\gamma^{(1)}, \ldots, \gamma^{(r)})$ and $\boldsymbol{\gamma}_{\prime} = (\gamma_{\prime}^{(1)}, \ldots, \gamma_{\prime}^{(r_{\prime})})$.

Consider the set of grid points

$$
\begin{aligned}
\mathbf{D}(M) := \{ \mathbf{u}_{\boldsymbol{\ell}^{(1)}} = (\ell_j^{(1)}/M_1)_{j=1,\ldots,r} \text{ and } \mathbf{v}_{\boldsymbol{\ell}^{(2)}} = (\ell_j^{(2)}/M_2)_{j=1,\ldots,r_{\prime}} \\
: \boldsymbol{\ell}^{(1)} = (\ell_1^{(1)}, \ldots, \ell_r^{(1)}) \in \{0, 1, \ldots, M_1\}^r, \\
\boldsymbol{\ell}^{(2)} = (\ell_1^{(2)}, \ldots, \ell_r^{(2)}) \in \{0, 1, \ldots, M_2\}^{r_{\prime}}, M_1 = M^{\widetilde{\beta}/\beta}, M_2 = M^{\widetilde{\beta}/\beta_{\prime}} \}.
\end{aligned}
$$

The cardinality of this set is $(M_1 + 1)^r \cdot (M_2 + 1)^{r_{\prime}}$. We write $\mathbf{u}_{\boldsymbol{\ell}^{(1)}} = (u_{\boldsymbol{\ell}^{(1)}}^{(j)})_{j=1,\ldots,r}$ and $\mathbf{v}_{\boldsymbol{\ell}^{(2)}} = (v_{\boldsymbol{\ell}^{(2)}}^{(j)})_{j=1,\ldots,r_{\prime}}$ to denote the components of $\mathbf{u}_{\boldsymbol{\ell}^{(1)}}$ and $\mathbf{v}_{\boldsymbol{\ell}^{(2)}}$ respectively. With slight abuse of notation we denote $\mathbf{w} = (\mathbf{u}, \mathbf{v}) = (u^{(1)}, \ldots, u^{(r)}, v^{(1)}, \ldots, v^{(r_{\prime})})$, $\boldsymbol{\ell} = (\boldsymbol{\ell}^{(1)}, \boldsymbol{\ell}^{(2)}) = (\ell_1^{(1)}, \ldots, \ell_r^{(1)}, \ell_1^{(2)}, \ldots, \ell_r^{(2)})$ and $\mathbf{w}_{\boldsymbol{\ell}} = (w_{\boldsymbol{\ell}}^{(j)})_{j=1,\ldots,r+r_{\prime}} = (\mathbf{u}_{\boldsymbol{\ell}^{(1)}}, \mathbf{v}_{\boldsymbol{\ell}^{(2)}}) = (u_{\boldsymbol{\ell}^{(1)}}^{(1)}, \ldots, u_{\boldsymbol{\ell}^{(1)}}^{(r)}, v_{\boldsymbol{\ell}^{(2)}}^{(1)}, \ldots, u_{\boldsymbol{\ell}^{(2)}}^{(r_{\prime})})$. Define

$$
\begin{aligned}
&P^{\beta,\beta_{\prime}} f(\mathbf{u}, \mathbf{v}) \\
&= P^{\beta,\beta_{\prime}} f(\mathbf{w}) \\
&:= \sum_{\mathbf{w}_{\boldsymbol{\ell}} \in \mathbf{D}(M)} P_{\mathbf{w}_{\boldsymbol{\ell}}}^{\beta,\beta_{\prime}} f(\mathbf{w}) \prod_{j=1}^{r+r_{\prime}} (1 - M_j |w^{(j)} - w_{\boldsymbol{\ell}}^{(j)}|)_+ \\
&= \sum_{\mathbf{u}_{\boldsymbol{\ell}^{(1)}}, \mathbf{v}_{\boldsymbol{\ell}^{(2)}} \in \mathbf{D}(M)} P_{\mathbf{u}_{\boldsymbol{\ell}^{(1)}}, \mathbf{v}_{\boldsymbol{\ell}^{(2)}}}^{\beta,\beta_{\prime}} f(\mathbf{u}, \mathbf{v}) \left( \prod_{j=1}^{r} (1 - M_1 |u^{(j)} - u_{\boldsymbol{\ell}^{(1)}}^{(j)}|)_+ \right) \left( \prod_{j=1}^{r_{\prime}} (1 - M_2 |v^{(j)} - v_{\boldsymbol{\ell}^{(2)}}^{(j)}|)_+ \right),
\end{aligned}
$$

where $M_j = M_1$ for $j = 1, \ldots, r$ and $M_j = M_2$ for $j = r+1, \ldots, r+r_{\prime}$.

**Lemma 5.** *If $f \in \mathcal{H}_{r,r_{\prime}}^{\beta,\beta_{\prime}}\left([0,1]^r,[0,1]^{r_{\prime}},K\right)$, then $\|P^{\beta,\beta_{\prime}}f - f\|_{L^\infty[0,1]^{r+r_{\prime}}} \leq KM^{-\widetilde{\beta}}$.*

*Proof.* Since for all $\mathbf{w} = (w^{(1)},\ldots,w^{(r+r_{\prime})}) \in [0,1]^{r+r_{\prime}}$,

$$\sum_{\mathbf{w}_{\boldsymbol{\ell}}\in\mathbf{D}(M)}\prod_{j=1}^{r+r_{\prime}}(1 - M_j|w^{(j)} - w_{\boldsymbol{\ell}}^{(j)}|)_+ = \prod_{j=1}^{r+r_{\prime}}\sum_{\ell=0}^{M_j}(1 - M_j|w^{(j)} - \ell/M_j|)_+ = 1, \tag{35}$$

we have

$$f(\mathbf{w}) = f(\mathbf{u},\mathbf{v})$$

$$= \sum_{\substack{\mathbf{u}_{\boldsymbol{\ell}(1)},\mathbf{v}_{\boldsymbol{\ell}(2)}\in\mathbf{D}(M):\\ \|\mathbf{u}-\mathbf{u}_{\boldsymbol{\ell}(1)}\|_\infty\leq 1/M_1\\ \|\mathbf{v}-\mathbf{v}_{\boldsymbol{\ell}(2)}\|_\infty\leq 1/M_2}} f(\mathbf{u},\mathbf{v})\left(\prod_{j=1}^{r}(1 - M_1|u^{(j)} - u_{\boldsymbol{\ell}(1)}^{(j)}|)_+\right)\left(\prod_{j=1}^{r_{\prime}}(1 - M_2|v^{(j)} - v_{\boldsymbol{\ell}(2)}^{(j)}|)_+\right)$$

and with (32),

$$\left|P^{\beta,\beta_{\prime}}f(\mathbf{u},\mathbf{v}) - f(\mathbf{u},\mathbf{v})\right| \leq \max_{\substack{\mathbf{u}_{\boldsymbol{\ell}(1)},\mathbf{v}_{\boldsymbol{\ell}(2)}\in\mathbf{D}(M):\\ \|\mathbf{u}-\mathbf{u}_{\boldsymbol{\ell}(1)}\|_\infty\leq 1/M_1\\ \|\mathbf{v}-\mathbf{v}_{\boldsymbol{\ell}(2)}\|_\infty\leq 1/M_2}}\left|P_{\mathbf{u}_{\boldsymbol{\ell}(1)},\mathbf{v}_{\boldsymbol{\ell}(2)}}^{\beta,\beta_{\prime}}f(\mathbf{u},\mathbf{v}) - f(\mathbf{u},\mathbf{v})\right|$$

$$\leq K\left(M_1^{-\beta} \vee M_2^{-\beta_{\prime}}\right) = K\,M^{-\widetilde{\beta}}.$$

$\square$

In the next few steps, we describe how to build a network that approximates $P^{\beta,\beta_{\prime}}f$.

**Lemma 6.** *Let $M, m$, be any positive integer. Denote $M_1 = M^{\widetilde{\beta}/\beta}$, $M_2 = M^{\widetilde{\beta}/\beta_{\prime}}$, $\mathsf{M} = (M_1+1)^r(M_2+1)^{r_{\prime}}$ and $\mathsf{r}_{\mathrm{sum}} = r + r_{\prime}$. Then there exists a network*

$$\mathrm{Hat}^{\mathsf{r}_{\mathrm{sum}}} \in \mathcal{F}\left(2 + (m+5)\lceil\log_2(\mathsf{r}_{\mathrm{sum}})\rceil, \mathsf{r}_{\mathrm{sum}}, 2\mathsf{r}_{\mathrm{sum}}\mathsf{M}, \mathsf{r}_{\mathrm{sum}}\mathsf{M}, 6\mathsf{r}_{\mathrm{sum}}\mathsf{M},\ldots,6\mathsf{r}_{\mathrm{sum}}\mathsf{M},\mathsf{M}), s, 1\right)$$

*with $s \leq 37\mathsf{r}_{\mathrm{sum}}^2\mathsf{M}(m+5)\lceil\log_2(\mathsf{r}_{\mathrm{sum}})\rceil$, such that $\mathrm{Hat}^r \in [0,1]^{\mathsf{M}}$ and for any $\mathbf{u} = (u^{(1)},\ldots,u^{(j)}) \in [0,1]^r$ and for any $\mathbf{v} = (v^{(1)},\ldots,v^{(j)}) \in [0,1]^{r_{\prime}}$*

$$\left|\mathrm{Hat}^{\mathsf{r}_{\mathrm{sum}}}(\mathbf{u},\mathbf{v}) - \left\{\left(\prod_{j=1}^{r}(1/M_1 - |u^{(j)} - u_{\boldsymbol{\ell}(1)}^{(j)}|)_+\right)\times\right.\right.$$

$$\left.\left.\left(\prod_{j=1}^{r_{\prime}}(1/M_2 - |v^{(j)} - v_{\boldsymbol{\ell}(2)}^{(j)}|)_+\right)\right\}_{\mathbf{u}_{\boldsymbol{\ell}(1)},\mathbf{v}_{\boldsymbol{\ell}(2)}\in\mathbf{D}(M)}\right|_\infty \leq \mathsf{r}_{\mathrm{sum}}^2 2^{-m}.$$

*For any $\mathbf{u}_{\boldsymbol{\ell}(1)},\mathbf{v}_{\boldsymbol{\ell}(2)} \in \mathbf{D}(M)$, the support of the function $(\mathbf{u},\mathbf{v}) \mapsto (\mathrm{Hat}^{r+r_{\prime}}(\mathbf{u},\mathbf{v}))_{\mathbf{u}_{\boldsymbol{\ell}(1)},\mathbf{v}_{\boldsymbol{\ell}(2)}}$ is moreover contained in the support of the function*

$$(\mathbf{u},\mathbf{v}) \mapsto \left\{\left(\prod_{j=1}^{r}(1/M - |u^{(j)} - u_{\boldsymbol{\ell}(1)}^{(j)}|)_+\right)\left(\prod_{j=1}^{r_{\prime}}(1/M - |v^{(j)} - v_{\boldsymbol{\ell}(2)}^{(j)}|)_+\right)\right\}.$$

*Proof.* **Step 1:** (For $r + r_{\prime} = 1$) Without loss of generality we consider the case when $r = 1$ and $r_{\prime} = 0$. We compute the functions $\{(u^{(j)} - \ell/M_1)_+\}_{j=1,\ell=0}^{r,M_1}$ and $\{(\ell/M_1 - u^{(j)})_+\}_{j=1,\ell=0}^{r,M_1}$ for the first hidden layer of the network. This requires $2r(M_1+1)$ units (nodes) and $2r(M_1+1)$ non-zero parameters.

For the second hidden layer we compute the functions $(1/M_1 - |u^{(j)} - \ell/M_1|)_+ = (1/M_1 - (u^{(j)} - \ell/M_1)_+ - (\ell/M_1 - u^{(j)})_+)_+$ using the output $(u^{(j)} - \ell/M_1)_+$ and $(\ell/M_1 - u^{(j)})_+$ from the output of the first hidden layer. This requires $r(M_1 + 1) + r_\prime(M_2 + 1)$ units (nodes) and $2r(M_1 + 1)$ non-zero parameters. This proves the result for the base case when $r + r_\prime = 1$.

**Step 2:** For $r + r_\prime > 1$, we compose the obtained network with networks that approximately compute the following

$$\left\{ \left( \prod_{j=1}^{r} (1/M_1 - |u^{(j)} - u_{\boldsymbol{\ell}^{(1)}}^{(j)}|)_+ \right) \left( \prod_{j=1}^{r_\prime} (1/M_2 - |v^{(j)} - v_{\boldsymbol{\ell}^{(2)}}^{(j)}|)_+ \right) \right\}_{\mathbf{u}_{\boldsymbol{\ell}^{(1)}}, \mathbf{v}_{\boldsymbol{\ell}^{(2)}} \in \mathbf{D}(M)}.$$

For fixed $\mathbf{u}_{\boldsymbol{\ell}^{(1)}}$ and $\mathbf{v}_{\boldsymbol{\ell}^{(2)}}$, and from the use of Lemma 8 there exist $\mathrm{Mult}_m^{r+r_\prime}$ networks in the class

$$\mathcal{F}\left(2 + (m+5)\lceil \log_2(r + r_\prime) \rceil, (r + r_\prime, 2(r + r_\prime), r + r_\prime, 6(r + r_\prime), 6(r + r_\prime), \dots, 6(r + r_\prime), 1)\right)$$

computing $(\prod_{j=1}^{r}(1/M_1 - |u^{(j)} - u_{\boldsymbol{\ell}^{(1)}}|)_+) \times (\prod_{j=1}^{r_\prime}(1/M_2 - |v^{(j)} - v_{\boldsymbol{\ell}^{(2)}}|)_+)$ up to an error that is bounded by $(r + r_\prime)^2 \, 2^{-m}$. Observe that we have two extra hidden layers to compute $(1/M_1 - |u^{(j)} - u_{\boldsymbol{\ell}^{(1)}}|)_+$ and $(1/M_2 - |v^{(j)} - v_{\boldsymbol{\ell}^{(2)}}|)_+$ for fixed $\mathbf{u}_{\boldsymbol{\ell}^{(1)}}$ and $\mathbf{v}_{\boldsymbol{\ell}^{(2)}}$ respectively, before we enter into the multinomial computation by regime invoking Lemma 8. Observe that the number of parameters in this network is upper bounded by $37(r + r_\prime)^2(m+5)\lceil \log_2(r + r_\prime) \rceil$.

Now we use the *parallelization* technique to have $(M_1 + 1)^r \cdot (M_1 + 1)^r$ parallel architecture for all elements of $\mathbf{D}(M)$. This provides the existence of the network with the number of non-zero parameters bounded by $37(r + r_\prime)^2(M_1 + 1)^r(M_2 + 1)^{r_\prime}(m+5)\lceil \log_2(r + r_\prime) \rceil$

By Lemma 8, for any $\mathbf{x} \in \mathbb{R}^r$, $\mathrm{Mult}_m^r(\mathbf{x}) = 0$ if one of the components of $\mathbf{x}$ is zero. This shows that for any $\mathbf{u}_{\boldsymbol{\ell}^{(1)}}, \mathbf{v}_{\boldsymbol{\ell}^{(2)}} \in \mathbf{D}(M)$, the support of the function $(\mathbf{u}, \mathbf{v}) \mapsto (\mathrm{Hat}^{r+r_\prime}(\mathbf{u}, \mathbf{v}))_{\mathbf{u}_{\boldsymbol{\ell}^{(1)}}, \mathbf{v}_{\boldsymbol{\ell}^{(2)}}}$ is contained in the support of the function $(\mathbf{u}, \mathbf{v}) \mapsto \left(\prod_{j=1}^{r}(1/M - |u^{(j)} - u_{\boldsymbol{\ell}^{(1)}}^{(j)}|)_+ \prod_{j=1}^{r_\prime}(1/M - |v^{(j)} - v_{\boldsymbol{\ell}^{(2)}}^{(j)}|)_+\right)$.

$\square$

*Proof of Theorem 5.* All the constructed networks in this proof are of the form $\mathcal{F}(L, \mathbf{p}, s) = \mathcal{F}(L, \mathbf{p}, s, \infty)$ with $F = \infty$. Denote $M_1 = M^{\tilde{\beta}/\beta}$, $M_2 = M^{\tilde{\beta}/\beta_\prime}$, $\beta_{\mathrm{sum}} = \beta + \beta_\prime$, and $r_{\mathrm{sum}} = r + r_\prime$. Let $M$ be the largest integer such that $\mathsf{M} = (M_1 + 1)^r(M_2 + 1)^{r_\prime} \leq N$ and define $L^* := (m+5)\lceil \log_2(\beta_{\mathrm{sum}} \vee r_{\mathrm{sum}}) \rceil$. Thanks to (34), (33) and Lemma 9, we can add one hidden layer to the network $\mathrm{Mon}_{m,\beta_{\mathrm{sum}}}^{r_{\mathrm{sum}}}$ to obtain a network

$$Q_1 \in \mathcal{F}\left(2 + L^*, (r, 6\lceil \beta \rceil C_{r_{\mathrm{sum}},\beta_{\mathrm{sum}}}, \dots, 6\lceil \beta \rceil C_{r_{\mathrm{sum}},\beta_{\mathrm{sum}}}, C_{r_{\mathrm{sum}},\beta_{\mathrm{sum}}}, \mathsf{M})\right),$$

such that $Q_1(\mathbf{u}, \mathbf{v}) \in [0, 1]^{\mathsf{M}}$ and for any $\mathbf{u} \in [0, 1]^r$ and for any $\mathbf{v} \in [0, 1]^{r_\prime}$

$$\left| Q_1(\mathbf{u}, \mathbf{v}) - \left( \frac{P^{\beta,\beta_\prime} f(\mathbf{u}, \mathbf{v})}{B} + \frac{1}{2} \right)_{\mathbf{u}_{\boldsymbol{\ell}^{(1)}}, \mathbf{v}_{\boldsymbol{\ell}^{(2)}} \in \mathbf{D}(M)} \right|_\infty \leq \beta_{\mathrm{sum}}^2 2^{-m} \tag{36}$$

with $B := \lceil 2K e^{r_{\mathrm{sum}}} \rceil$. The total number of non-zero parameters in the $Q_1$ network is $6r_{\mathrm{sum}}(\beta_{\mathrm{sum}} + 1)C_{r_{\mathrm{sum}},\beta_{\mathrm{sum}}} + 42(\beta_{\mathrm{sum}} + 1)^2 C_{r_{\mathrm{sum}},\beta_{\mathrm{sum}}}^2(L^* + 1) + C_{r_{\mathrm{sum}},\beta_{\mathrm{sum}}}\mathsf{M}$.

Recall that the network $\mathrm{Hat}^{r_{\mathrm{sum}}}$ computes the products of hat functions (splines) $(\prod_{j=1}^{r}(1/M_1 - |u^{(j)} - u_{\boldsymbol{\ell}^{(1)}}|)_+)(\prod_{j=1}^{r_\prime}(1/M_2 - |v^{(j)} - v_{\boldsymbol{\ell}^{(2)}}|)_+)$ up to an error that is bounded by $r_{\mathrm{sum}}^2 2^{-m}$. It requires at most $37 r_{\mathrm{sum}}^2 N L^*$ active parameters. Observe that $C_{r_{\mathrm{sum}},\beta_{\mathrm{sum}}} \leq (\beta_{\mathrm{sum}} + 1)^{r_{\mathrm{sum}}} \leq N$ by the definition of $C_{r,\beta}$ and the assumptions on $N$. By Lemma 6, the networks $Q_1$ and $\mathrm{Hat}^{r_{\mathrm{sum}}}$ can be embedded into a joint parallel network $(Q_1, \mathrm{Hat}^{r_{\mathrm{sum}}})$ with $2 + L^*$ hidden layers of size $(r_{\mathrm{sum}}, 6(r_{\mathrm{sum}} + \lceil \beta_{\mathrm{sum}} \rceil)N, \dots, 6(r_{\mathrm{sum}} + \lceil \beta_{\mathrm{sum}} \rceil)N, 2\mathsf{M})$. Using $C_{r,\beta} \vee (M + 1)^r \leq N$ again, the number of non-zero parameters in the combined network $(Q_1, \mathrm{Hat}^r)$ is bounded by

$$6r_{\mathrm{sum}}(\beta_{\mathrm{sum}} + 1)C_{r_{\mathrm{sum}},\beta_{\mathrm{sum}}} + 42(\beta_{\mathrm{sum}} + 1)^2 C_{r_{\mathrm{sum}},\beta_{\mathrm{sum}}}^2(L^* + 1) + C_{r_{\mathrm{sum}},\beta_{\mathrm{sum}}}\mathsf{M} + 37 r_{\mathrm{sum}}^2 N L^*$$
$$\leq 42(r_{\mathrm{sum}} + \beta_{\mathrm{sum}} + 1)^2 C_{r_{\mathrm{sum}},\beta_{\mathrm{sum}}} N(1 + L^*) \tag{37}$$
$$\leq 84(r_{\mathrm{sum}} + \beta_{\mathrm{sum}} + 1)^{3 + r_{\mathrm{sum}}} N(m + 5),$$

where for the last inequality, we used $C_{r_{\mathrm{sum}},\beta_{\mathrm{sum}}} \leq (\beta_{\mathrm{sum}} + 1)^{r_{\mathrm{sum}}}$, the definition of $L^*$ and that for any $x \geq 1$, $1 + \lceil \log_2(x) \rceil \leq 2 + \log_2(x) \leq 2(1 + \log(x)) \leq 2x$.

Next, we pair the $(\mathbf{u}_{\boldsymbol{\ell}(1)}, \mathbf{v}_{\boldsymbol{\ell}(2)})$-th entry of the output of $Q_1$ and $\mathrm{Hat}^r$ and apply to each of the M pairs the $\mathrm{Mult}_m$ network described in Lemma 7. In the last layer, we add all entries. By Lemma 7 this requires at most $24(m+5)\mathsf{M}+\mathsf{M} \leq 25(m+5)N$ active parameters for the M multiplications and the sum. Using Lemma 7, Lemma 6, (36) and triangle inequality, there exists a network $Q_2 \in \mathcal{F}(2 + L^* + m + 6, (r_{\mathrm{sum}}, 6(r_{\mathrm{sum}} + \lceil\beta_{\mathrm{sum}}\rceil)N, \ldots, 6(r_{\mathrm{sum}} + \lceil\beta_{\mathrm{sum}}\rceil)N, 1))$ such that for any $\mathbf{u} \in [0,1]^r$ and for any $\mathbf{v} \in [0,1]^{r'}$

$$\left| Q_2(\mathbf{u}, \mathbf{v}) - \sum_{\mathbf{u}_{\boldsymbol{\ell}(1)}, \mathbf{v}_{\boldsymbol{\ell}(2)} \in \mathbf{D}(M)} \left( \frac{P^{\beta,\beta'} f(\mathbf{u}, \mathbf{v})}{B} + \frac{1}{2} \right) \left( \prod_{j=1}^{r} (1/M_1 - |u^{(j)} - u^{(j)}_{\boldsymbol{\ell}(1)}|)_+ \right) \right.$$
$$\left. \left( \prod_{j=1}^{r'} (1/M_2 - |v^{(j)} - v^{(j)}_{\boldsymbol{\ell}(2)}|)_+ \right) \right|$$

$$\leq \sum_{\substack{\mathbf{u}_{\boldsymbol{\ell}(1)}, \mathbf{v}_{\boldsymbol{\ell}(2)} \in \mathbf{D}(M): \\ \|\mathbf{u}-\mathbf{u}_{\boldsymbol{\ell}(1)}\|_\infty \leq 1/M_1 \\ \|\mathbf{v}-\mathbf{v}_{\boldsymbol{\ell}(2)}\|_\infty \leq 1/M_2}} (1 + r^2_{\mathrm{sum}} + \beta^2_{\mathrm{sum}}) 2^{-m}$$

$$\leq (1 + r^2_{\mathrm{sum}} + \beta^2_{\mathrm{sum}}) 2^{r-m}. \tag{38}$$

Here, the first inequality follows from the fact that the support of $(\mathrm{Hat}^{r+r'}(\mathbf{u}, \mathbf{v}))_{\mathbf{u}_{\boldsymbol{\ell}(1)}, \mathbf{v}_{\boldsymbol{\ell}(2)}}$ is contained in the support of $\left( \prod_{j=1}^{r} (1/M - |u^{(j)} - u^{(j)}_{\boldsymbol{\ell}(1)}|)_+ \prod_{j=1}^{r'} (1/M - |v^{(j)} - v^{(j)}_{\boldsymbol{\ell}(2)}|)_+ \right)$ (see Lemma 6). Because of (37), the network $Q_2$ has at most

$$109(r_{\mathrm{sum}} + \beta_{\mathrm{sum}} + 1)^{3+r_{\mathrm{sum}}} N(m + 5) \tag{39}$$

non-zero parameters.

To obtain a network reconstruction of the function $f$, it remains to scale and shift the output entries. This is not entirely trivial because of the bounded parameter weights in the network. Recall that $B = \lceil 2Ke^r \rceil$. The network $x \mapsto BM_1^r M_2^{r'} x$ is in the class $\mathcal{F}(3, (1, M_1^r M_2^{r'}, 1, \lceil 2Ke^r \rceil, 1))$ with shift vectors $\mathbf{v}_j$ are all equal to zero and weight matrices $W_j$ with all entries equal to one. Because of $N \geq (K+1)e^{r_{\mathrm{sum}}}$, the number of parameters of this network is bounded by $2M_1^r M_2^{r'} + 2\lceil 2Ke^r \rceil \leq 6N$. This shows existence of a network in the class $\mathcal{F}(4, (1, 2, 2M_1^r M_2^{r'}, 2, 2\lceil 2Ke^r \rceil, 1))$ computing $a \mapsto BM_1^r M_2^{r'}(a - c)$ with $c := 1/(2M_1^r M_2^{r'})$. This network computes in the first hidden layer $(a - c)_+$ and $(c - a)_+$ and then applies the network $x \mapsto BM_1^r M_2^{r'} x$ to both units. In the output layer, the second value is subtracted from the first one. This requires at most $6 + 12N$ active parameters.

Because of (38) and (35), there exists a network $Q_3$ in

$$\mathcal{F}\big((m + 13) + L^*, (r_{\mathrm{sum}}, 6(r_{\mathrm{sum}} + \lceil\beta_{\mathrm{sum}}\rceil)N, \ldots, 6(r_{\mathrm{sum}} + \lceil\beta_{\mathrm{sum}}\rceil)N, 1)\big)$$

such that

$$\left| Q_3(\mathbf{u}, \mathbf{v}) - \sum_{\mathbf{u}_{\boldsymbol{\ell}(1)}, \mathbf{v}_{\boldsymbol{\ell}(2)} \in \mathbf{D}(M)} P^{\beta,\beta'} f(\mathbf{u}, \mathbf{v}) \left( \prod_{j=1}^{r} (1/M_1 - |u^{(j)} - u^{(j)}_{\boldsymbol{\ell}(1)}|)_+ \right) \right.$$
$$\left. \left( \prod_{j=1}^{r'} (1/M_2 - |v^{(j)} - v^{(j)}_{\boldsymbol{\ell}(2)}|)_+ \right) \right|$$

$$\leq (2K + 1)M_1^r M_2^{r'}(1 + r^2_{\mathrm{sum}} + \beta^2_{\mathrm{sum}})(2e)^{r_{\mathrm{sum}}} 2^{-m}, \quad \text{for all } (\mathbf{u}, \mathbf{v}) \in [0,1]^{r_{\mathrm{sum}}}.$$

With (39), the number of non-zero parameters of $Q_3$ is bounded by

$$109(\mathsf{r}_{\text{sum}} + \beta_{\text{sum}} + 1)^{3+\mathsf{r}_{\text{sum}}} N(m+6).$$

Observe that by construction $\mathsf{M} = (M_1 + 1)^r (M_2 + 1)^{r'} \leq N \leq (3M_1)^r (3M_2)^{r'} = 3^{\mathsf{r}_{\text{sum}}} M^{\widetilde{r}}$ and hence $M^{-\widetilde{\beta}} \leq N^{-\widetilde{\beta}/\widetilde{r}} 3^{\mathsf{r}_{\text{sum}} \widetilde{\beta}/\widetilde{r}}$. Together with Lemma 5, the result follows. $\qquad\square$

### L.1. Embedding properties of neural network function classes

We denote $\mathcal{F}(L, \boldsymbol{p})$ as the class of neural networks with $L$ hidden layers and $\boldsymbol{p} \in \mathbb{N}^{L+2}$ nodes per layer. The class $\mathcal{F}(L, \boldsymbol{p})$ is subset of $\mathcal{F}(L, \boldsymbol{p})$ with the sparsity parameter $s$.

For the approximation of a function by a network, we first construct smaller networks computing simpler objects. Let $\mathbf{p} = (p_0, \ldots, p_{L+1})$ and $\mathbf{p}' = (p'_0, \ldots, p'_{L+1})$. To combine networks, we make frequent use of the following rules.

*Enlarging:* $\mathcal{F}(L, \mathbf{p}, s) \subseteq \mathcal{F}(L, \mathbf{q}, s')$ whenever $\mathbf{p} \leq \mathbf{q}$ componentwise and $s \leq s'$.

*Composition:* Suppose that $f \in \mathcal{F}(L, \mathbf{p})$ and $g \in \mathcal{F}(L', \mathbf{p}')$ with $p_{L+1} = p'_0$. For a vector $\mathbf{v} \in \mathbb{R}^{p_{L+1}}$ we define the composed network $g \circ \sigma_{\mathbf{v}}(f)$ which is in the space $\mathcal{F}(L + L' + 1, (\mathbf{p}, p'_1, \ldots, p'_{L'+1}))$. In most of the cases that we consider, the output of the first network is non-negative and the shift vector $\mathbf{v}$ will be taken to be zero.

*Additional layers/depth synchronization:* To synchronize the number of hidden layers for two networks, we can add additional layers with an identity weight matrix, such that

$$\mathcal{F}(L, \mathbf{p}, s) \subset \mathcal{F}(L + q, (\underbrace{p_0, \ldots, p_0}_{q \text{ times}}, \mathbf{p}), s + qp_0). \tag{40}$$

*Parallelization:* Suppose that $f, g$ are two networks with the same number of hidden layers and the same input dimension, that is, $f \in \mathcal{F}(L, \mathbf{p})$ and $g \in \mathcal{F}(L, \mathbf{p}')$ with $p_0 = p'_0$. The parallelized network $(f, g)$ computes $f$ and $g$ simultaneously in a joint network in the class $\mathcal{F}(L, (p_0, p_1 + p'_1, \ldots, p_{L+1} + p'_{L+1}))$.

### L.2. Technical lemmas for the proof of Theorem 5

We use $\mathcal{F}(L, \mathbf{r})$ to denote a fully connected network with $L$ deep layers and $\mathbf{r} \in \mathbb{N}_0^{L+2}$ representing the nodes in each layer.

The following technical lemmas are required for the proof of Theorem 5. Lemma 7, Lemma 8, and Lemma 9 restate Lemma A.2, Lemma A.3, and Lemma A.4 from (Schmidt-Hieber, 2020), respectively.

**Lemma 7.** *For any positive integer $m$, there exists a network* $\text{Mult}_m \in \mathcal{F}(m + 4, (2, 6, 6, \ldots, 6, 1))$, *such that* $\text{Mult}_m(x, y) \in [0, 1]$,

$$\big| \text{Mult}_m(x, y) - xy \big| \leq 2^{-m}, \quad \text{for all } x, y \in [0, 1],$$

*and* $\text{Mult}_m(0, y) = \text{Mult}_m(x, 0) = 0$.

**Lemma 8.** *For any positive integer $m$, there exists a network*

$$\text{Mult}_m^r \in \mathcal{F}((m + 5)\lceil \log_2 r \rceil, (r, 6r, 6r, \ldots, 6r, 1))$$

*such that* $\text{Mult}_m^r \in [0, 1]$ *and*

$$\left| \text{Mult}_m^r(\mathbf{x}) - \prod_{i=1}^r x_i \right| \leq r^2 2^{-m}, \quad \text{for all } \mathbf{x} = (x_1, \ldots, x_r) \in [0, 1]^r.$$

*Moreover,* $\text{Mult}_m^r(\mathbf{x}) = 0$ *if one of the components of $\mathbf{x}$ is zero.*

The number of monomials with degree $|\boldsymbol{\alpha}| < \gamma$ is denoted by $C_{r,\gamma}$. Obviously, $C_{r,\gamma} \leq (\gamma + 1)^r$ since each $\alpha_i$ has to take values in $\{0, 1, \ldots, \lfloor \gamma \rfloor\}$.

**Lemma 9.** *For $\gamma > 0$ and any positive integer $m$, there exists a network*

$$\mathrm{Mon}_{m,\gamma}^r \in \mathcal{F}\big(1 + (m+5)\lceil \log_2(\gamma \vee 1)\rceil, (r, 6\lceil \gamma \rceil C_{r,\gamma}, \ldots, 6\lceil \gamma \rceil C_{r,\gamma}, C_{r,\gamma})\big),$$

*such that $\mathrm{Mon}_{m,\gamma}^r \in [0,1]^{C_{r,\gamma}}$ and*

$$\left| \mathrm{Mon}_{m,\gamma}^r(\mathbf{x}) - (\mathbf{x}^{\boldsymbol{\alpha}})_{|\boldsymbol{\alpha}|<\gamma} \right|_\infty \leq \gamma^2 2^{-m}, \quad \textit{for all } \mathbf{x} \in [0,1]^r.$$

