# OpenReview forum: "A Likelihood Based Approach to Distribution Regression Using Conditional  Deep Generative Models"
_ICML.cc/2025/Conference — ICML 2025 poster_

### Official Review · Reviewer_vpHs · 2025-03-11

**Overall Recommendation:** 3

**Summary:**

The paper investigates a likelihood-based framework for regression in a view of conditional distribution estimation. The authors propose a Sieve Maximum Likelihood Estimator (MLE) to estimate the conditional density of the response given predictors. Importantly, they derive convergence rates (under Hellinger and Wasserstein metrics) that depend solely on the intrinsic dimension and smoothness of the true conditional distribution. Specifically, by incorporating a small noise perturbation into the data distribution, it shows to improve the estimation. This insight is validated with numerical experiments on both synthetic datasets and MNIST data, illustrating that the proposed likelihood-based approach outperforms several existing methods

**Claims And Evidence:**

The paper clearly establishes that the convergence rates for the proposed sieve MLE depend only on the intrinsic dimension and smoothness of the true conditional distribution. The derivations that motivate adding a small noise perturbation to handle nearly singular distributions are mathematically sound. The empirical explorations are limited mostly to synthetic examples and MNIST data. More diverse real-world experiments would help convince the reader that the approach reliably circumvents high-dimensional challenges. Moreover, the theoretical analysis suggests that injecting a small noise is crucial when the data are very close to a lower-dimensional manifold. Although some experiments illustrate performance improvements with noise perturbation, it is unclear how to choose the magnitude of the injected noise, which is not fully explored.

**Essential References Not Discussed:**

CARD is a method motivated to address regression problems in a view of conditional generative models. It also provides several synthetic and real-world empirical justifications, which should be considered to compare to enhance the soundness of this paper.

Han, X., Zheng, H., & Zhou, M. (2022). Card: Classification and regression diffusion models. Advances in Neural Information Processing Systems, 35, 18100-18115.

**Experimental Designs Or Analyses:**

I reviewed the experimental designs and analyses described in the paper. Here are the key points along with some issues:

- Most experiments use synthetic data. While these controlled experiments support the theoretical claims, more extensive testing on diverse real-world datasets would help demonstrate the method’s robustness and practical utility.

- The experimental setup does not include an in-depth analysis of how variations in key hyperparameters (network depth, sparsity, noise injection magnitude) affect performance. Understanding these sensitivities would be beneficial, especially since the theoretical convergence rates depend on noise and smoothness parameters.

- The evaluation is mainly based on MSE and Wasserstein distance. Additional evaluations, such as those used in uncertainty estimation, and additional baseline methods to compare with, could provide further insights into the performance of the conditional density estimates.

**Methods And Evaluation Criteria:**

The use of sieve MLE and the derivation of convergence rates in both the Hellinger and Wasserstein metrics directly address the statistical challenges of estimating nearly singular conditional distributions. This design effectively tackles the curse of dimensionality by focusing on intrinsic dimensions for conditional generative models.
The convergence rates in Hellinger and Wasserstein distances are used and are appropriate for assessing the quality of conditional density estimation. These metrics capture both the pointwise differences and the overall transport cost between the estimated and true distributions, which is critical when dealing with singular supports and avoid the potential issues posed in KL divergence. To this end, both the methods and evaluation criteria are thoughtfully chosen to match the problem’s demands.

**Other Comments Or Suggestions:**

Please see the weakness above.

**Other Strengths And Weaknesses:**

The font of text and the plotted shapes in Figure 2 and 3 seem to be distorted, making the figure hard to read.

**Questions For Authors:**

Please see the above sections.

**Relation To Broader Scientific Literature:**

The work builds on deep generative models and related likelihood-based approaches to estimate complex conditional distributions. This approach leverages a conditional generator modeled by deep neural networks to capture the data’s structure.

The derivation of convergence rates in both the Hellinger and Wasserstein metrics connects with a rich body of work in optimal transport. This naturually formalizes the benefits of small noise injection in learning nearly singular distributions, which has been practically used in generative modeling, such as training GANs with additive noise (e.g., with diffusion schedule), training autoregressive models with noisy distribution.

By analyzing deep generative models with composite structures and considering both sparse and fully connected network classes, the paper generalizes earlier findings on deep network approximation capabilities. This demonstrates that the framework can adapt to a broad class of conditional distributions, including those that are nearly singular, which is a significant extension over standard density estimation methods.

**Theoretical Claims:**

The proofs for the key theoretical claims are largely correct and built on well-established methods, though they rely on strong assumptions.
Moreover, the smoothness theorem is based on earlier work in the literature. While the authors’ application of these results appears appropriate, a more detailed verification could reveal sensitivities regarding constant factors or the exact rate dependencies.

---

> ### Author Rebuttal · Authors · 2025-04-01
>
> We thank the reviewer for their insightful and constructive feedback, and we sincerely appreciate their recognition of the strengths in our likelihood‐based framework and its theoretical underpinnings.
>
> **Claims and Evidence:**  We thank the Reviewer for highlighting these key aspects of our theoretical analysis, the primary focus of our work. The simulations and MNIST experiments are designed to validate our statistical insights rather than introduce a new algorithm. Moreover, our framework is broad and equally applicable to other likelihood‐based methods—such as Normalizing Flows—as demonstrated in related empirical works [A] and [B].
>
> Our theory indicates that a small noise perturbation is needed to smooth nearly singular data distributions. This is demonstrated with our synthetic manifold example and further supported by additional simulations (please see end of Rebuttal 1) which show how noise levels affect the MNIST W1 distance. In theory, this noise level is guided by the data distribution properties, and practically often chosen via cross validation as explored in [I], balancing minimal distortion with stable optimization
>
> **Theoretical Claims:**  While it is possible to derive the exact constants in our proofs, doing so can be cumbersome and distract from the main insights. Nevertheless, we will make every effort to track the constants that are tied to key problem characteristics. Our focus remains on clearly illustrating how these constants depend on critical variables—such as the intrinsic dimension and smoothness parameters—while remaining independent of the sample size n. As is common in the literature (e.g., [J, K]), tight upper bounds suffice to capture convergence behavior without specifying exact numerical values.
>
> We consider our assumptions to be reasonably mild. In particular:
> - Assumption 1 guarantees a density exists for both noisy and noiseless data on the manifold—which is widely acceptable and foundational in statistics and machine learning.
> - Assumptions 2 and 3 are practically motivated. For example, [C] provides empirical evidence for lower-dimensional data structures, and Assumption 2 generalizes common structures like multiplicative and additive models [M].
>
> **Experimental Designs Or Analyses:**
> - Our experiments with synthetic data and MNIST illustrate and validate the key theoretical insights of our approach. While further empirical validation on diverse real-world datasets and architectures would better demonstrate robustness and practical utility (see [A, B]), a comprehensive evaluation is beyond our current scope. We expect these results to generalize to more complex image datasets, as benchmark data typically have intrinsic dimensionality on the order of 10¹ [L].
> - We have added quantitative analysis on MNIST to assess how the W1 distance varies with different noise levels (please see at the end of Rebuttal 1). Additionally, we evaluated sample size effects by comparing performance at 25%, 50%, and 75% of the data. In full-dimensional cases (FD1, FD2, FD3) where smoothness is critical, performance improves significantly with increased sample size. Although our current experiments show minimal variations with network depth and sparsity, we acknowledge their importance and plan to explore these further.
> - Since our framework is built on Wasserstein and Hellinger distances, our analysis focuses on these metrics. Nonetheless, exploring additional evaluations—such as uncertainty quantification and further baseline comparisons—would provide valuable insights. We view this as a promising direction for future work aimed at supporting optimal theoretical rates for these additional approaches.
>
> **Essential References Not Discussed**: We shall include Han, Zheng, and Zhou (2022) in our references. While CARD employs a score‐based, empirically focused approach, our work establishes nearly tight convergence rates for likelihood‐based methods (including VAEs and normalizing flows) in conditional density estimation, offering new insights into overcoming the curse of dimensionality.
>
> **Other Strengths And Weaknesses:** We have improved the resolution and clarity of Figures 2 and 3, ensuring clear fonts and shapes. These enhancements will be in the final manuscript, subject to permission.
>
> We hope these explanations, along with the additional quantitative analyses, address your concerns and further strengthen our manuscript. We welcome any additional questions or further discussions.
>
> **References:** Please see at the end of Rebuttal 2.

---

> > ### Comment · Reviewer_vpHs · 2025-04-03
> >
> > I appreciate the authors' responses. I will keep my score.

---

> > > ### Author Response · Authors · 2025-04-09
> > >
> > > Dear Reviewer vpHs,
> > >
> > > As the discussion period nears its conclusion, we would like to sincerely thank you for your thoughtful review and for taking the time to consider our responses.
> > >
> > > We have done our best to address all of your concerns and questions in our rebuttal. We acknowledge your decision to maintain your score.
> > >
> > > If there are any remaining points of clarification or concerns, we would be happy to address them within the remaining time.
> > >
> > > Thank you again for your time and consideration.
> > >
> > > Best regards,
> > >
> > > The Authors

---

### Official Review · Reviewer_8PjD · 2025-03-12

**Overall Recommendation:** 3

**Summary:**

In this paper, the authors explore the theoretical properties of deep neural network-based generative models for conditional density estimation. They tackle the specific problem of estimating a conditional distribution on a lower dimensional manifold, and find the convergence rate to be dependent on the intrinsic dimension and smoothness of the true conditional distribution. These results provide theoretical support for why deep generative models are effective at learning densities on a manifold in high dimensional space, and why noising data on the manifold may be beneficial.

## update after rebuttal
I appreciate the authors' further clarifications and simplifications. Taking into consideration the authors' response to my and the other reviews, and that this is primarily a theoretical work, I have updated my score accordingly, recommending weak accept.

**Claims And Evidence:**

The paper claims 1) to be the first to explore the convergence properties of deep generative models for conditional density estimation in a low-d manifold, 2) as a result, provide theoretical justification for why deep generative models have been empirically successful in modeling real world data in such scenarios (e.g., images), and 3) numerically demonstrate these claims in synthetic datasets and MNIST.

I do not know the literature well enough to evaluate 1) or 2). In terms of 3, quantitative results on the synthetic task seem satisfactory (Table 1), but evaluation on the “real world” dataset of MNIST is rather qualitative. In particular, samples look quite blurry and lack diversity. It’s not that the qualitative performance is underwhelming, it’s more that I don’t quite understand what these results show—that the VAE setup with sparse / fully connected networks *can* model such low-d embedded distributions?

Aside from that, I was not able to understand the paper well enough, specifically Section 2, to evaluate the theoretical results fully.

**Essential References Not Discussed:**

This work seems quite relevant: Normalizing flows for estimating conditional densities on a manifold: https://arxiv.org/abs/2003.13913

Not necessarily a problem for the current contributions, but definitely in the same problem space, and therefore should be discussed.

**Experimental Designs Or Analyses:**

The synthetic and MNIST datasets seem suitable to demonstrate the points. On the other hand, it would have been nice to see slightly more complex datasets and architectures.

**Methods And Evaluation Criteria:**

I’m not sure whether the classes of neural networks considered here (sparse and FC feedforward) are complex enough to empirically support the theoretical points, since most modern DGMs come with more inductive biases (when working with real data) that are explicitly designed to first reduce the data dimensionality, e.g., from pixel space to an embedding via convolution. Additionally, as mentioned above, I did not find the visual evaluation on MNIST convincing.

**Other Comments Or Suggestions:**

Perhaps a high-level outline of what Section 2 aims to achieve, step by step, would be informative.

**Other Strengths And Weaknesses:**

Not necessarily the authors’ fault, and potentially more so my lack of expertise, but I found the theoretical results, and especially the notations, to be very difficult to follow along.

**Questions For Authors:**

- how would the results generalize to (or are related to) deep architectures closer to those used in the last few years, e.g., flows, GANs, diffusion models, as the authors initially cite?
- can the results on MNIST be quantified?
- do the results hold for datasets more complex and/or relevant than MNIST?

**Relation To Broader Scientific Literature:**

A theoretical understanding of how and why modern deep generative models work is an important research direction, given its prevalence in modern machine learning. That being said, I was not able to evaluate how meaningful of a contribution the current work is.

**Theoretical Claims:**

Was not able to evaluate.

---

> ### Author Rebuttal · Authors · 2025-04-01
>
> We sincerely thank the reviewer for their thoughtful and constructive feedback. While our theoretical contributions are inherently technical, we have made every effort to present them in a manner accessible to a broader audience. Below, we address each of the reviewer’s points in detail:
>
> **Claims And Evidence**: Our work establishes nearly tight convergence rates for conditional density estimation on low-dimensional manifolds by analyzing deep generative models. The proofs throughout the paper follow a unified strategy that addresses two primary sources of error. First, we bound the statistical (estimation) error using entropy bounds (Eq. 10 and 11). Second, we quantify the approximation error—the bias incurred when approximating the true generator with our network class. Theorem 1 provides a general convergence rate, while Corollary 1 specializes the result to sparse and fully connected architectures, directly linking the rate to network parameters (depth, width, and sparsity). This adaptive behavior lets the estimator depend primarily on the intrinsic dimension and smoothness of the true model, mitigating the curse of dimensionality.
>
> For nearly singular distributions, where data lie extremely close to a low-dimensional manifold—Corollary 2 demonstrates that adding a small noise perturbation smoothens the gaps. This noise injection smooths the data distribution and enables a favorable Wasserstein convergence rate, thus extending the ideas from Theorem 2. In essence, the noise helps overcome challenges inherent in estimating distributions that do not have full-dimensional support.
>
> **Extension to More General Settings**: Theorem 3 broadens our analysis to generators with differing smoothness across dimensions. By combining our new approximation result (Theorem 5) with modified entropy bounds, we show that the overall convergence rate depends on a combined measure of smoothness and intrinsic dimension, reinforcing the robustness and adaptability of our framework
>
> **Methods And Evaluation Criteria**: We acknowledge the reviewer’s concerns regarding the empirical evaluation. Our focus is primarily theoretical; the experiments were chosen to validate the statistical insights of our approach. Although MNIST is a high-dimensional dataset (D = 784), it naturally exhibits a low intrinsic dimension [C]. The presented simulations and MNIST experiments illustrate key trends predicted by our theory. Moreover, the framework we develop is general and applies equally to other likelihood-based methods, such as Normalizing Flows, as highlighted in related works [A] and [B].
>
> **Questions For Authors**:
> - While our theoretical analysis is carried out for sparse and fully connected feedforward networks, our tools are sufficiently general to extend to more advanced architectures—such as convolutional networks, GANs, and diffusion models. Our approximation-theoretic tools (e.g., Theorem 5) are potentially applicable to these architectures [F, G]. Future work will aim to incorporate these more complex models, further bridging the gap between theory and state-of-the-art empirical performance.
> - Additional quantitative MNIST experiment results have been added (please see end of Rebuttal 1).
> - For more complex image datasets, we expect these results to generalize as empirical evidence suggests that benchmark datasets typically exhibit intrinsic dimensionality on the order of $10^1$ [C].
>
> **Essential References and Other Weaknesses**: We appreciate the reviewer’s suggestions. In the revised manuscript, we will include the recommended reference (arxiv:2003.13913). Regarding notations, most are standard in the literature (e.g., [F, G, H]).
>
> We hope that these clarifications, together with the additional analyses, address your concerns and further strengthen our manuscript. Should you have any further questions or require additional details, we are happy to discuss them further.
>
> **All Rebuttal References**
>
> - [A] Conditional variational autoencoder for neural machine translation. arXiv:1812.04405.
> - [B] Learning likelihoods with conditional normalizing flows. arXiv:1912.00042.
> - [C] The intrinsic dimension of images and its impact on learning. arXiv:2104.08894.
> - [D] Vector diffusion maps and the connection Laplacian. arXiv:1102.0075.
> - [E] Estimating the reach of a manifold. arXiv:1705.04565.
> - [F] Diffusion models are minimax optimal distribution estimators. arXiv:2303.01861.
> - [G] Adaptivity of diffusion models to manifold structures. PMLR:238.
> - [H] Smooth function approximation by deep neural networks. arXiv:1906.06903.
> - [I] A likelihood approach to nonparametric estimation of a singular distribution. arXiv:2105.04046.
> - [J] Statistical theory for image classification using deep convolutional neural networks. arXiv:2011.13602
> - [K] Nonparametric regression using deep neural networks. arXiv:1708.06633.
> - [L] Generalized Additive Models: Some Applications (JASA, 1987)

---

### Official Review · Reviewer_p9g3 · 2025-03-24

**Overall Recommendation:** 3

**Summary:**

This submitted manuscript works on the statistical aspects of conditional generative models, which is an important topic in the broad area of deep generative models. The analysis focuses on the likelihood-based models and, under the assumption that the high dimensional response variables have low-dimensional data manifold, studies the convergence rate of a sieve MLE. The finding, as stated in the paper, tries to explain why deep generative models can overcome the curse of dimensionality.

In the experimental section, the authors demonstrate the performance in terms of relevant theorems presented in the paper.

**Claims And Evidence:**

There are three major assumptions that have been made. The first one is about the generative process from the underlying true distribution following the canonical form in Equation 2. The second one is about the composite function class. The last one builds the identifiability condition.

The Thm1 is followable, and the proof for Thm2 in the appendix is also understandable. But I am having trouble fully understanding the rest of the proofs and Corollaries.

**Essential References Not Discussed:**

Overall, the paper is well written and cited all the key papers in the related areas.

**Experimental Designs Or Analyses:**

There is one concern that I have about the experiments. As the paper mainly focuses on the statistical theory part of deep conditional generative models, the experimental data seems to be trivial. It is either from simulation or low-dimensional real data. And the authors claim that the statistical analysis reveals why the conditional deep generative models can avoid the curse of dimensionality. But in the experimental section, there is no dataset that has high dimensional data. Overall, I am a bit concerned how the statistical analysis helps in reality.

**Methods And Evaluation Criteria:**

The authors demonstrate their theoretical analysis through simulation data and real-world datasets. There is a quantitative and qualitative illustration that shows the performance in relevance to the statistical bounds. As I am not from the background of statistical theory, I cannot fully verify the appropriateness of the methods used to analyze the conditional deep generative models.

**Other Comments Or Suggestions:**

Although it is a well-written paper, I find it would be better to include a starting toy example to demonstrate the idea.

**Other Strengths And Weaknesses:**

The paper is quite theoretical and well-written in terms of problem setup and proofs. It links several key papers in the field of generative models and the story naturally follows. However, the weakness is that the experimental sections lack of convincing evidence to support the theoretical findings. Another minor weakness is that the assumption 3 excluding the interior point seems a bit too strong. I am not sure if it holds in reality.

**Questions For Authors:**

1) Do you observe a similar pattern if you experiment on large-resolution image datasets?

**Relation To Broader Scientific Literature:**

I am fully capable of judging the novelty in terms of statistical theory aspects. From my reading through the paper, it does broaden my view of deep generative models from the statistical point of view. However, the concern arises from the experimental section where the scale of the dataset is limited.

**Theoretical Claims:**

As I point out previously, my background is limited to judging the novelty and rigorous of the the theoretical claims presented in the paper. But by following the proof in the appendix, I can understand most of the proofs.

---

> ### Author Rebuttal · Authors · 2025-04-01
>
> We thank the reviewer for their detailed and constructive feedback, and we appreciate the positive remarks on the main results and the comprehensibility of our proofs.
>
> **Our derivations beyond Theorems 1 and 2** follow a unified strategy controlling two sources of error. First, we bound the statistical (estimation) error using entropy bounds (Eq. 10 and 11), which essentially capture our function class’s complexity via covering or bracketing numbers. Second, we quantify the approximation error—the bias incurred when approximating the true generator with our network class. Theorem 1 establishes a general convergence rate by balancing these errors, while Corollary 1 specializes the result to sparse and fully connected network architectures. With appropriate tuning parameters such as depth, width, and sparsity, the estimator adapts to the intrinsic dimension and smoothness of the true model, crucial for mitigating the curse of dimensionality.
>
> For nearly singular distributions, where data concentrate near a low-dimensional manifold, Corollary 2 demonstrates that a small noise perturbation smooths distribution gaps, achieving a favorable Wasserstein convergence rate and extending Theorem 2. Theorem 3 further expands our analysis to generators with varying smoothness across dimensions. By combining our new approximation result (Theorem 5) with adjusted entropy bounds, we established that the convergence rate in these settings depends on a composite measure of smoothness and intrinsic dimension.
>
> **Methods And Evaluation Criteria and Experimental Designs Or Analyses**: We would like to emphasize that our work is primarily theoretical. The simulations and MNIST experiments serve to validate our statistical insights rather than introduce a new algorithm. While some experiments use low-dimensional data in the experiments, MNIST (D = 784) is high-dimensional yet exhibits low intrinsic dimensionality (around 10). Additionally, our framework is broad and applies equally to other likelihood-based methods—such as Normalizing Flows—as demonstrated in related empirical works [A] and [B].
>
> **Other Comments**: We appreciate the suggestion. In the revised manuscript, we will add a toy example—using a one-dimensional generator map map that transforms a uniform variable into one with a β-Hölder smooth density: $u\mapsto u^{1/(\beta+1)}$. Setting $\beta^*$ = 1 and $t^*$ = 1 in our framework reinterprets our results in this concrete setting, clearly illustrating the theoretical ideas.
>
> **Other Strengths And Weaknesses**: Assumption 3 holds in practice, ensuring the manifold’s closure is well defined—a common requirement in manifold learning [E]—and is crucial for maintaining low-dimensional structure. If the manifold were space-filling in $\mathbb{R}^D$, the data would lose their low-dimensional properties. The concept of reach, related to the manifold’s condition number, is well established in the literature [C, D, E] and ensures our analysis remains robust even when the support is singular.
>
> We hope these clarifications, along with the additional analyses (see below) and proposed examples, address your concerns and strengthen our manuscript. We welcome any further questions or discussion.
>
> **Additional Experiment Addressing All Rebuttal Feedback**
>
> To address concerns about the empirical quality of our MNIST images, we computed the W1 distance metric (W1 ± SD). For each digit, we averaged the W1 distances over 50 images and aggregated the results. For context, the average distance between two test images—reflecting baseline variability—is 2.0219 ± 0.745. Below is a table summarizing the empirical results (noise denotes the $\sigma$ of $N(0,\sigma^2)$ added to the training data):
> |Noise|Sparsely Connected|Fully Connected|
> |-|-|-|
> |0|1.9555 ± 0.7182|1.9555 ± 0.7182|
> |0.005|1.9478 ± 0.7329|1.8663 ± 0.6251|
> |0.01|1.9503 ± 0.7291|1.9598 ± 0.6867|
> |0.02|2.0699 ± 0.6937|2.0616 ± 0.7410|
> |0.04|2.2199 ± 0.6735|2.2117 ± 0.6627|
> |0.06|2.3487 ± 0.6576|2.3172 ± 0.6267|
> |0.08| 2.4623 ± 0.6245| 2.4076 ± 0.6308|
> |0.1|2.5734 ± 0.6492| 2.5002 ± 0.6337|
> |0.3|3.4931 ± 0.7012| 3.4943 ± 0.7164|
> |0.5|4.0880 ± 0.7518| 4.0995 ± 0.7633|
>
> At zero noise, both architectures yield W1 distances slightly lower than the baseline, indicating that the generated samples align well with the test images. As noise increases, the W1 distance grows steadily, showing that higher noise degrades image quality by pushing samples further from the true distribution. Both network types exhibit similar performance across noise levels, suggesting robustness to architecture choice, although subtle differences hint at variations in noise sensitivity.
> While extracting precise convergence rates from these experiments is challenging, the observed trends validate the large-sample properties for manifold-supported data and align with our theoretical expectations.
>
> We propose including this analysis in the appendix alongside our MNIST experiments.

---

### Decision · Program_Chairs · 2025-05-01

**Decision:**

Accept (poster)

**Comment:**

The paper considers conditional generative likelihood models in high dimensional problems where the data lives on a low dimensional manifold. It presents several theoretical results to analyze the proposed method and some basic experiments to validate their model. Three reviewers all leaned towards accepting the paper, but were not strong in their confidence. While they expressed some lack of confidence in their ability to evaluate the theoretical contributions, these seem correct and well-developed from the standards of ICML. The main concern was with the empirical evaluation, which focused more on demonstrating the proposed method than how it compares with others and are not stress tested, for example the data considered is essentially toy data and not the realistic high dimensional data that the paper claims to care about. However, considering the impressive theoretical content, the paper can contribute to the conference by providing insight on the analytical properties of conditional generative models, which would make the paper worth accepting.